# PDX1+ cell budding morphogenesis in a stem cell-derived islet spheroid system

Jia Zhao [1,10] ✉, Shenghui Liang[1,10], Haoning Howard Cen[1], Yanjun Li[2], Robert K. Baker[1], Balwinder Ruprai[1], Guang Gao[3], Chloe Zhang[1], Huixia Ren [2,4], Chao Tang [4], Liangyi Chen [2,5], Yanmei Liu [6,7], Francis C. Lynn [8], James D. Johnson [1] & Timothy J. Kieffer [1,9] ✉

Remarkable advances in protocol development have been achieved to manufacture insulin-secreting islets from human pluripotent stem cells (hPSCs). Distinct from current approaches, we devised a tunable strategy to generate islet spheroids enriched for major islet cell types by incorporating PDX1+ cell budding morphogenesis into staged differentiation. In this process that appears to mimic normal islet morphogenesis, the differentiating islet spheroids organize with endocrine cells that are intermingled or arranged in a core-mantle architecture, accompanied with functional heterogeneity. Through in vitro modelling of human pancreas development, we illustrate the importance of *PDX1* and the requirement for EphB3/4 signaling in eliciting cell budding morphogenesis. Using this new approach, we model Mitchell-Riley syndrome with *RFX6* knockout hPSCs illustrating unexpected morphogenesis defects in the differentiation towards islet cells. The tunable differentiation system and stem cell-derived islet models described in this work may facilitate addressing fundamental questions in islet biology and probing human pancreas diseases.

In type 1 diabetes (T1D), beta cells in pancreatic islets are selectively targeted for autoimmune destruction, leading to a marked deficiency in insulin secretion[1–3]. Islet transplantation has been proven to be an effective cell replacement therapy for T1D[4–6], but widespread use of this procedure is restricted by scarcity of available donors and the need for chronic immunosuppression. Sufficient quantities of insulin-producing islets can be obtained by in vitro step-wise differentiation of human pluripotent stem cells (hPSCs) through mimicking the process of fetal pancreas development[7–10]. Moreover, such protocols can model mechanisms of pancreas diseases, test therapies, and enable the discovery of drugs to promote islet (re)generation. Despite many

protocol advancements[11–15], shortcomings remain with differentiation consistency and functional competency of the resulting insulin-producing cells.

Efforts to develop stem cell-derived islets typically focus on recreating key transcriptional and chromatin landscapes by modulation of biochemical signaling pathways involved in pancreatic development[14,15]. However, the developing pancreas is also exposed to a time-course of defined morphogenesis, which renders a locally heterogeneous niche, with mechanical forces and morphogen gradients to instruct asynchronous differentiation, islet formation and proper cell function[16,17]. This is somewhat overlooked in present hPSC

[1]Life Sciences Institute, Departments of Cellular & Physiological Sciences and Surgery, University of British Columbia, Vancouver, BC, Canada. [2]Institute of Molecular Medicine, School of Future Technology, National Biomedical Imaging Center, Peking University, Beijing, China. [3]Imaging Core Facility, Life Sciences Institute, University of British Columbia, Vancouver, BC, Canada. [4]Center for Quantitative Biology, Peking University, Beijing, China. [5]PKU-IDG/McGovern Institute for Brain Research, Peking University, Beijing, China. [6]Key Laboratory of Brain, Cognition and Education Sciences, Ministry of Education, South China Normal University, 510631 Guangzhou, China. [7]Institute for Brain Research and Rehabilitation, and Guangdong Key Laboratory of Mental Health and Cognitive Science, South China Normal University, 510631 Guangzhou, China. [8]BC Children's Hospital Research Institute, Department of Surgery, University of British Columbia, Vancouver, BC, Canada. [9]School of Biomedical Engineering, University of British Columbia, Vancouver, BC, Canada. [10]These authors contributed equally: Jia Zhao, Shenghui Liang. ✉e-mail: jia.zhao@ubc.ca; tim.kieffer@ubc.ca

differentiation strategies and may contribute to the immaturity of the resulting islet cells. Although there are differences in the timing of transcriptional regulation between species, the phases and key developmental events of pancreas development are highly conserved[17–22]. During early embryogenesis, the human fetal pancreas undergoes specification of a presumptive pancreatic region marked by expression of *PDX1*, demarcation of pancreatic domains in the ventral and dorsal foregut, bud formation, and outgrowth (lumenogenesis, branching morphogenesis), followed by segregation of tip and trunk regions, endocrine differentiation with a single wave of *NGN3* expression and islet morphogenesis (endocrine cell clustering, cytoarchitectural remodeling, islet vascularization and innervation)[20,21]. Modeling the complex process of pancreatic morphogenesis with differentiating hPSCs may inform the design of islet cell therapies with optimal function and cellular organization.

In this study, we developed a stem cell-derived human islet spheroid system which involves multiple morphological changes during the staged differentiation. We show that islets form through a budding process mediated by early progenitor cell sorting. Correlated with functional heterogeneity, these human islet spheroids may adopt an intermingled or core-mantle cytoarchitecture that can be patterned in vitro by defined morphogen signals and antioxidants. Using knockout hPSC lines, we further demonstrate that this system enables modeling of pancreas diseases and probing disease mechanisms with human cells in a dish. The knowledge gained from this work informs strategies of incorporating morphogenesis cues for further optimizing islet production from hPSCs as well as provides unique morphological insights into understanding fundamental questions of islet development and explaining disease phenotypes.

## Results

### Development of a budding-type stem cell-derived islet spheroid system

We and others previously developed protocols for bulk differentiation of hPSCs into hormone-expressing islet cells by timed addition of defined soluble factors[8–13,23–25]. This bulk-type differentiation starts from a robust induction of definitive endoderm (DE) cells through activation of TGFβ and Wnt signaling. By fine tuning the doses of Wnt agonists at the DE stage (Stage 1), we established a budding-type differentiation system generating islet cells that are highly enriched in bud structures (Fig. 1a and Supplementary Fig. 1). Specifically, depending on the cell lines used, budding-type differentiation was induced with CHIR99021 concentrations of 0.2–1.5 μM (or MCX-928 at 0.1–0.5 μM; Wnt^low) whereas bulk-type differentiation was induced (i.e., PDX1+ buds are no longer formed) when CHIR99021 was used at 3 μM (or 1 μM MCX-928; Wnt^med) in combination with GDF8 or Activin A (Supplementary Fig. 1a–c). In the presence of TGFβ ligand, medium-level Wnt activation (Wnt^med) was sufficient to drive hPSCs to differentiate into DE cells (>95% FOXA2+/SOX17+ cells) whereas low-level Wnt activation (Wnt^low) generated ~75% FOXA2+/SOX17+ DE cells within Stage 1 cell populations (Supplementary Fig. 1a, d–f), suggesting that the extent of Wnt signaling (as examined by >60 Wnt pathway genes, Supplementary Fig. 2) determined endoderm specification efficiency. In the Wnt^med bulk-type differentiation, an average of 95% PDX1+ cells and 73% PDX1+/NKX6.1+ pancreatic progenitors were generated in Stage 4 cultures, and insulin was expressed throughout Stage 7 clusters (Supplementary Fig. 1b–d, g). The Wnt^low budding-type differentiation produced an average of 32% PDX1+ cells and ~ 5% PDX1+/NKX6.1+ cells at the end of Stage 4, and insulin was largely restricted to a local area of Stage 7 clusters (Supplementary Fig. 1b, c, e, g), indicating that the single-day treatment (for Stage 1 day 1 only) with Wnt^med or Wnt^low condition drastically impacted subsequent differentiation toward the endocrine lineage.

By comparing transcriptomic differences in cells generated by the two protocols (Supplementary Fig. 3a), we find that pancreatic cell

transcripts (*PDX1, NKX6.1, SOX9*) and pro-endocrine cell transcripts (*NGN3, NEUROD1*) were less abundant in budding-type Stage 4 cells (Supplementary Fig. 3b). The expression levels of most transcripts became comparable between budding-type and bulk-type cells at Stages 5 and 7 (Supplementary Fig. 3c, d). Nevertheless, relative to bulk-type cells, we note higher levels of *GHRL, SLC18A1, KCNK1* transcripts and lower levels of *GCG, PPY, ARX, NKX2.2, ABCC8* transcripts in budding-type Stage 7 cells relative to cells from bulk-type differentiations (Supplementary Fig. 3d). Although transcripts of functional beta cell markers *MAFA, IAPP, GCK, PSCK1, KCNK3* were similar between the two types of differentiation (Supplementary Fig. 3d), insulin secretion and total insulin content of budding-type Stage 7 beta cells was 1.4-fold and 1.7-fold lower than those of bulk-type cells, respectively (Supplementary Fig. 1h), suggesting a less mature phenotype of beta cells generated by the budding protocol. As initial seeding densities were the same and total cell numbers were indistinguishable between the two types of differentiation (Supplementary Fig. 1i), these differences are unlikely attributed to seeding condition, distinct cell survival or growth rate during early-stage culture.

To examine whether the budding-type differentiation could be reproduced with different Wnt inducers and in multiple hPSC lines, three Wnt agonists (CHIR99021, MCX-928 and mWnt3a) were tested and four hPSC lines (one hiPSC and three hESC lines) were differentiated under Wnt^low conditions (Supplementary Fig. 4). When combined with TGFβ ligands, all the three Wnt agonists used at low concentrations induced a budding-type differentiation process, while increasing the concentrations of CHIR99021 and MCX-928 to a medium level resulted in a switch to a more uniform bulk-type differentiation (Supplementary Fig. 4a). Consistent across the four hPSC lines, 70–85% FOXA2+/SOX17+ DE cells at the end of Stage 1 and 20–40% PDX1+ cells at the end of Stage 4 were obtained from the Wnt^low-mediated budding-type differentiation (Supplementary Fig. 4b–d). Dithizone staining of Stage 7 clusters showed zinc-enriched beta cells in bud structures (Supplementary Fig. 4e), whose cells appeared to be more compact and had a darker appearance than main bodies under phase contrast imaging (Supplementary Fig. 4f), again supporting the concept that the Wnt^low conditions used at the DE stage induce a budding-type differentiation in vitro.

Next, we quantified the efficiencies of bud formation (including PDX1+ pancreatic buds and INS+ islet buds) during the differentiation process with our Wnt^low protocol on various hPSC lines (Supplementary Fig. 5). The percentages of clusters that formed PDX1+ bud(s) for the cell lines examined were 98.7% ± 0.2% (HUES4 PDXeG), 93.1% ± 3.1% (Mel1 *INS^GFP/W*), 88.3% ± 0.9% (H1 hESC), 87.7% ± 1.4% (GCaMP hiPSC) and 97.3% ± 0.4% (HUES8) (Supplementary Fig. 5a, b, d). The percentages of clusters that formed INS+ islet bud(s) quantified at the end of protocol were 92.5% ± 4.1% (HUES4 PDXeG), 85.4% ± 3.2% (Mel1 *INS^GFP/W*), 77.4% ± 10.2% (H1 hESC), 71.4% ± 12.3% (GCaMP hiPSC) and 84.5% ± 5.9% (HUES8) (Supplementary Fig. 5e, f, h). The number of PDX1+ buds or INS+ islet buds per main body was also quantified using HUES4 PDXeG and Mel1 *INS^GFP/W* reporter lines, both showing >75% main bodies with only one bud and ~10% main bodies with two or more buds (Supplementary Fig. 5c, g). These results demonstrate the high efficiency and reproducibility of bud formation observed with the Wnt^low protocol in various hPSC lines.

### Islet formation through budding morphogenesis is mediated by PDX1+ cell clustering

To characterize the budding process in further detail, we used Mel1 *INS^GFP/W* hESCs (an insulin reporter line[26]) to monitor the differentiation process. Consistently, PDX1 was expressed in about one-third of Stage 4 cells generated with the budding-type differentiation protocol (Fig. 1b). After single-cell dissociation and aggregation of Stage 4 cells into clusters followed by transition to static suspension culture for

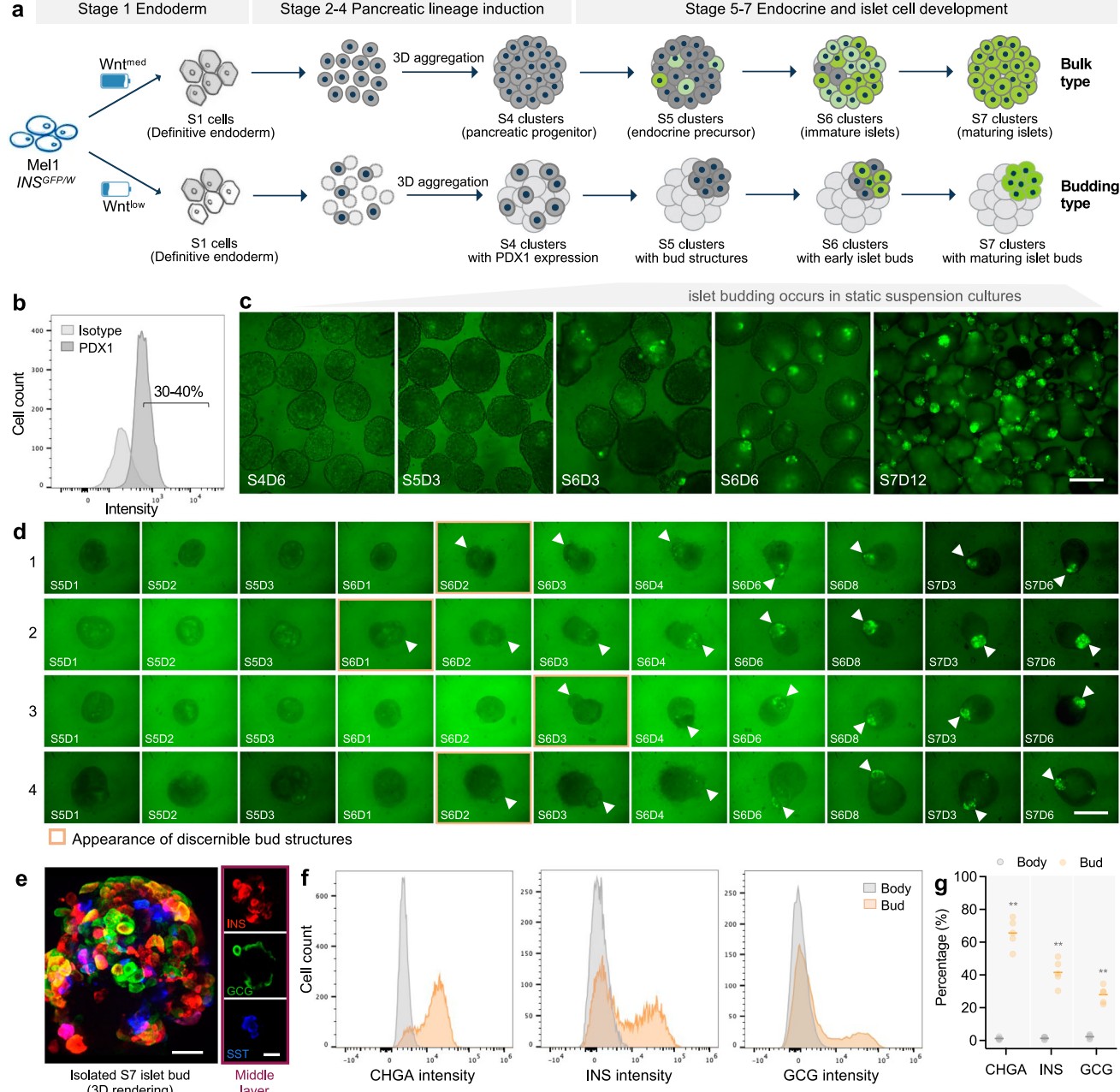

**Fig. 1 | Development of a budding-type hPSC-derived islet spheroid system.**
**a** Schematic of a tunable differentiation strategy to generate two types of islet models from stem cells under static suspension culture. The bulk- or budding-type differentiation is established by simply fine tuning the doses of Wnt agonist at DE stage. Specifically, Wnt^med condition at DE stage induces bulk type whereas Wnt^low condition at DE stage induces budding type. The "bulk" and "budding" are defined as to whether islet cells are broadly developed throughout clusters or enriched in local area of clusters. Created with BioRender.com released under a Creative Commons Attribution-NonCommercial-NoDerivs 4.0 International license. **b** Typical flow cytometry plot of Stage 4 cells from budding differentiation cultures. **c** Live imaging showing that a small proportion of cells were committed into INS+ cells (indicated by INS-GFP reporter). Of note, INS+ cells were enriched in small bud-like structures rather dispersed in entire clusters over the course of differentiation. Scale bar, 300 μm. **d** Tracking individual cluster in living cultures shows that the appearance of budding structures preceded the gradual induction of INS+ cells within the buds. Scale bar, 300 μm. **e** Representative image of Stage 7 islet bud stained for major islet cell types. INS insulin, GCG glucagon, SST somatostatin. Scale bars, 20 μm. Representative flow plots (**f**) and quantification (**g**) of CHGA+, INS+, and GCG+ cells within the bud and body compartments after enzymatic isolation (see *Methods*) from Stage 7 clusters. *n* = 5 independent differentiations, \*\**p* < 0.01 versus body, unpaired two-tailed t-test. Budding differentiation was induced by 100 ng/mL GDF8 plus 1–1.5 μM CHIR99021 for Stage 1 Day 1 using the Mel1 *INS*^GFP/W^ line. Source data are provided as a Source Data file.

further endocrine cell induction, we noted that a proportion of cells were committed to insulin-positive cells as indicated by expression of the GFP reporter. Interestingly, these insulin-expressing cells were enriched within bud-like structures rather than dispersed throughout clusters (Fig. 1c). To track this process in living cultures, we differentiated individual clusters in ultralow attachment U-bottom 96-wells.

Live imaging revealed that bud structures appeared during Stage 5 to early Stage 6 protruding from main body clusters, followed by a gradual induction of insulin-positive cells within the buds (Fig. 1d). Additional characterization showed that these buds were primarily endocrine cells and enriched in all major islet cell types (Fig. 1e–g and Supplementary Fig. 6).

Since the formation of bud structures preceded induction of insulin-positive cells (Fig. 1d), we sought to characterize these initial bud structures through expression of progenitor markers. Based upon

observations that insulin-positive cells were specifically derived in the bud niche associated with elevated expression of PDX1 (Fig. 2a), we reasoned that formation of initial bud structures was a result of PDX1+

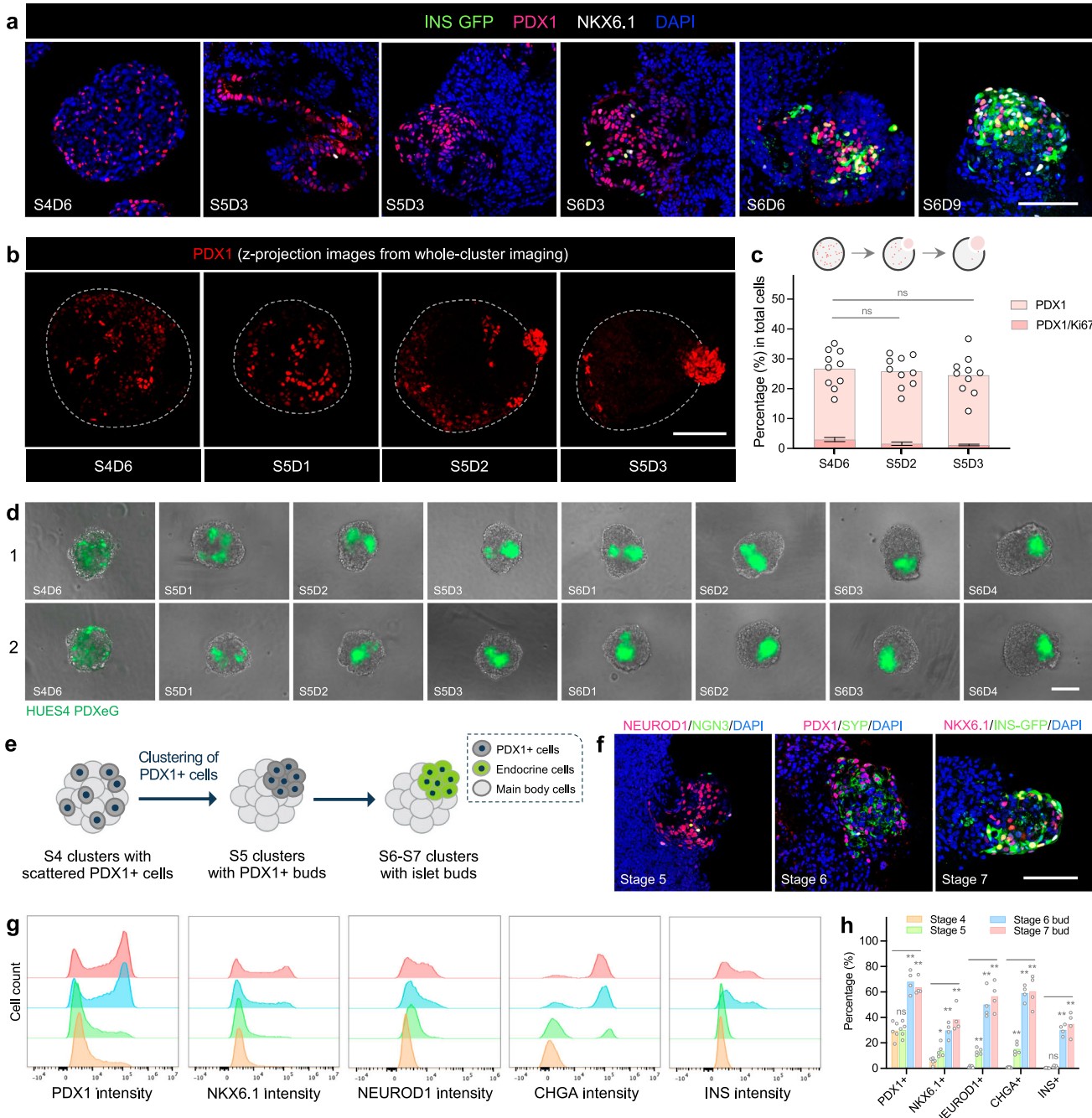

**Fig. 2 | Clustering of PDX1+ cells initiates bud niche formation and facilitates local endocrine cell induction. a** Representative images of Stage 4-6 clusters stained for PDX1 and NKX6.1. INS expression was indicated by INS-GFP in clusters derived from Mel1 $INS^{GFP/W}$ hESCs. Scale bar, 50 μm. **b** Representative z-projection images of Stage 4-5 clusters (highlighted with white dotted lines) stained for PDX1 from whole-cluster imaging. Scale bar, 100 μm. **c** Quantification of the percentage of PDX1+ cells and proliferative PDX1+ cells at indicated stages. $n = 3$–10 independent differentiations, ns not significant, one-way ANOVA with Dunnett test for multiple comparisons to S4D6 clusters. Created with BioRender.com released under a Creative Commons Attribution-NonCommercial-NoDerivs 4.0 International license. **d** Daily monitoring of individual cluster showing the process of PDX1+ cell clustering. A *PDX1* reporter hESC line (HUES4 PDXeG) was used for the time-course snapshot visualization. Scale bar, 100 μm. **e** Schematic of PDX1+ cell

clustering and proposed consequences on following differentiation. Created with BioRender.com released under a Creative Commons Attribution-NonCommercial-NoDerivs 4.0 International license. **f** Representative images of Stage 5-7 clusters stained for PDX1, NKX6.1, and various endocrine markers. NGN3 neurogenin 3. SYP synaptophysin. Scale bar, 50 μm. Representative flow cytometry plots (**g**) and quantification (**h**) of Stage 4-5 entire clusters and Stage 6-7 isolated buds examined for pancreatic progenitor markers (PDX1, NKX6.1), endocrine markers (CHGA, NEUROD1) and INS expression. $n = 4$–5 independent differentiations, ns not significant, *$p < 0.05$, **$p < 0.01$, one-way ANOVA with Dunnett test for multiple comparisons to Stage 4 clusters. Budding differentiation was induced by 100 ng/mL GDF8 plus 1–1.5 μM CHIR99021 for Stage 1 Day 1 using the Mel1 $INS^{GFP/W}$ line (**a–c, f–h**) and HUES4 PDXeG line (**d**). Source data are provided as a Source Data file.

cell enrichment, either through cell migration or local proliferation of PDX1+ cells. With whole-cluster imaging, we observed a progressive change from an initial scattering of PDX1+ cells throughout entire clusters towards the high enrichment in bud structures, suggesting a migration of PDX1+ cells from main bodies to buds (Fig. 2b). The proportions of PDX1+ cells remained consistent along with the low PDX1+ cell proliferation rate (Fig. 2c and Supplementary Fig. 7) during budding process indicating that formation of PDX1+ cell-enriched buds is unlikely due to local expansion of PDX1+ cells within the buds. Finally, to directly visualize PDX1+ cells in living cultures, we used a *PDX1* reporter hESC line (HUES4 PDXeG[27]) while imaging the process of bud formation (Supplementary Fig. 8a). We noted PDX1+ cells tended to form clustered structures in both planar and suspension cultures (Supplementary Fig. 8b–i) and the clustering process appeared to be independent of differentiation factors in culture media (Supplementary Fig. 8j). By tracking PDXeG-positive cells in individual clusters over the course of differentiation with time-course snapshot imaging, we demonstrated migration of PDX1+ cells forming the initial bud structures protruding from main bodies (Fig. 2d).

The PDX1+ cell-enriched bud niche may facilitate local endocrine cell induction (Fig. 2e). In support of this hypothesis, we detected expression of pro-endocrine markers NEUROD1 and NGN3 (transient and sporadic) in Stage 5 bud structures, pan-endocrine marker synaptophysin (SYN) in Stage 6 islet buds, as well as expression of PDX1 and NKX6.1 in INS+ cells of Stage 7 islet buds (Fig. 2a, f). Similarly, flow analysis of entire clusters from Stages 4-5 and isolated islet buds from Stages 6-7 showed sequential expression of PDX1, NKX6.1, NEUROD1, CHGA, and INS over the course of budding-type differentiation (Fig. 2g, h), suggesting that endocrine cell development within bud structures followed a progression similar to that observed in bulk-type differentiation of hPSCs.

## Cytoarchitectural rearrangement and functional heterogeneity of differentiating islet buds

Budding-type differentiations produced islet buds comprised of major islet cell types including insulin-expressing beta cells, glucagon-expressing alpha cells and somatostatin-expressing delta cells (Fig. 1e–g). Interestingly, we observed the majority of Stage 7 buds organized into a core-mantle architecture with beta cells in the center and alpha cells in the periphery (Fig. 3a, b). Quantification of islet cell composition revealed that Stage 6 islet buds contained mostly INS +/GCG+ bi-hormonal cells, whereas Stage 7 islet buds had significantly increased proportions of INS+/GCG- and GCG+/INS- monohormonal cells (Fig. 3c). To explore whether biochemical cues affected these changes, we examined the effects of either adding or subtracting key components, which are unique in either Stage 6 or Stage 7 medium, in an extended culture following Stage 6 (Supplementary Fig. 9a). Immunostaining showed that the extended culture promoted a transition to monohormonal cells for all conditions, except the basal medium condition (Supplementary Fig. 9b). Cytoarchitecturally, removal or inclusion of Stage 6-unique compounds (LDN, GSiXX) did not change the intermingled organization of islet cells; by contrast, inclusion of Stage 7-unique compounds (R428, NAC, Trolox) in extended cultures appeared to induce the core-mantle structure (Supplementary Fig. 9b, c). However, by individually supplementing Stage 7-unique components we did not see effects on cytoarchitectural changes (Supplementary Fig. 9), suggesting that there is a synergistic requirement for more than one factor, rather than a single biochemical cue tested here, for the islet cell rearrangement.

We next examined the function of developing islet buds by utilizing a microfluidic chip-based perifusion system[28] (termed "microperifusion") to perfuse single clusters with simultaneous imaging of intracellular calcium levels (Fig. 3d and Supplementary Fig. 10a). Robust insulin secretion of individual primary human islets from different donors was consistently detected

(Supplementary Fig. 10b). With the setup validated, we first performed calcium imaging on islet buds and categorized calcium activities in response to 3.3 mM and 16.7 mM glucose with an unsupervised clustering algorithm[29]. Three distinct responsive types were identified in Stage 6 buds, with calcium activities synchronized among cells within certain responsive types but unsynchronized among different responsive types. In Stage 7 buds a single cluster type was frequently identified, suggesting that the cells displayed generally synchronized calcium responses in the bud (Fig. 3e and Supplementary Movies 1, 2). Quantification revealed that Stage 6 islet buds were identified with more calcium-response types (4.4 ± 1.4 versus 1.7 ± 0.2 in Stage 7 buds) while Stage 7 islet buds had more synchronized cells per active region (16.2 ± 2.9 versus 6.9 ± 1.4 in Stage 6 buds) (Fig. 3f, g), in keeping with the presence of more homotypic interactions among beta cells in Stage 7 buds. Next, by perfusing individual clusters for insulin secretion measurement we observed high glucose (HG)-responders, low glucose (LG)-responders, and glucose non-responders in both Stage 6-7 islet buds, albeit at different proportions (Fig. 3h, i). LG-responders were more common in Stage 6 buds while HG-responders were observed more frequently in Stage 7 buds. Although all spheroids were responsive to depolarization with KCl, the majority were not responsive to elevated glucose concentrations, indicating the developing islet buds were functionally immature (Fig. 3h, i). To correlate function with cytoarchitecture, we retrieved individual islet buds from chips after microperifusion for immunostaining. The HG-responders in Stage 6-7 islet buds mainly adopted a core-mantle organization with predominant monohormonal islet cells, while LG-responders in Stage 6-7 buds had a mixed organization predominantly consisting of bi-hormonal INS+/GCG+ cells (Fig. 3i), supporting the concept that monohormonal INS+ cells are more faithfully responsive to glucose challenges relative to bi-hormonal cells[9].

## Modeling human pancreas diseases with the tunable differentiation system

The transcription factor *PDX1* is a master regulator of pancreas lineage commitment[30], with *PDX1* deficiency causing pancreatic agenesis seen in various models[31–33]. Consistent with this, we observed a requirement for *PDX1* in eliciting cell budding morphogenesis during pancreatic patterning by applying both bulk- and budding-type differentiations with a *PDX1* knockout (KO) hESC line[32]. The absence of *PDX1* did not impact DE formation, but stopped formation of pancreatic progenitors, endocrine cells and major islet cell types as well as insulin production (Fig. 4 and Supplementary Fig. 11). Moreover, while pancreatic buds were consistently formed in wildtype (WT) hESC-derived clusters with our budding differentiation protocol, cell budding (or pancreatic bud formation) was blocked in differentiating clusters derived from the *PDX1* KO hESCs regardless of differentiation pipeline applied (Fig. 4e and Supplementary Fig. 11a, b). These results corroborate previous work by recapitulating the pancreatic agenesis phenotype with a stem cell model and again emphasize the importance of *PDX1* in pancreatic specification and morphogenesis.

To further explore the utility of our in vitro system in modeling other pancreatic diseases, we examined the impact of *RFX6* deficiency using a *RFX6* KO hESC line[32], given *RFX6* gene mutations can cause pancreas hypoplasia[34]. *RFX6* is required for pancreatic progenitor differentiation and maintenance of mature alpha and beta cell function[34–37]. However, it is presently unknown whether *RFX6* directly affects early pancreatic morphogenesis. To address this, we assessed the impact of *RFX6* deficiency using both differentiation pipelines. While *RFX6* KO did not affect DE specification, it significantly compromised the induction of PDX1+ cells (KO: 44.3% ± 1.2% versus WT: 94.1% ± 3.8%) and PDX1+/NKX6.1+ pancreatic progenitors (KO: 36% ± 0.9% versus WT: 75.5% ± 2.1%) in our bulk differentiation

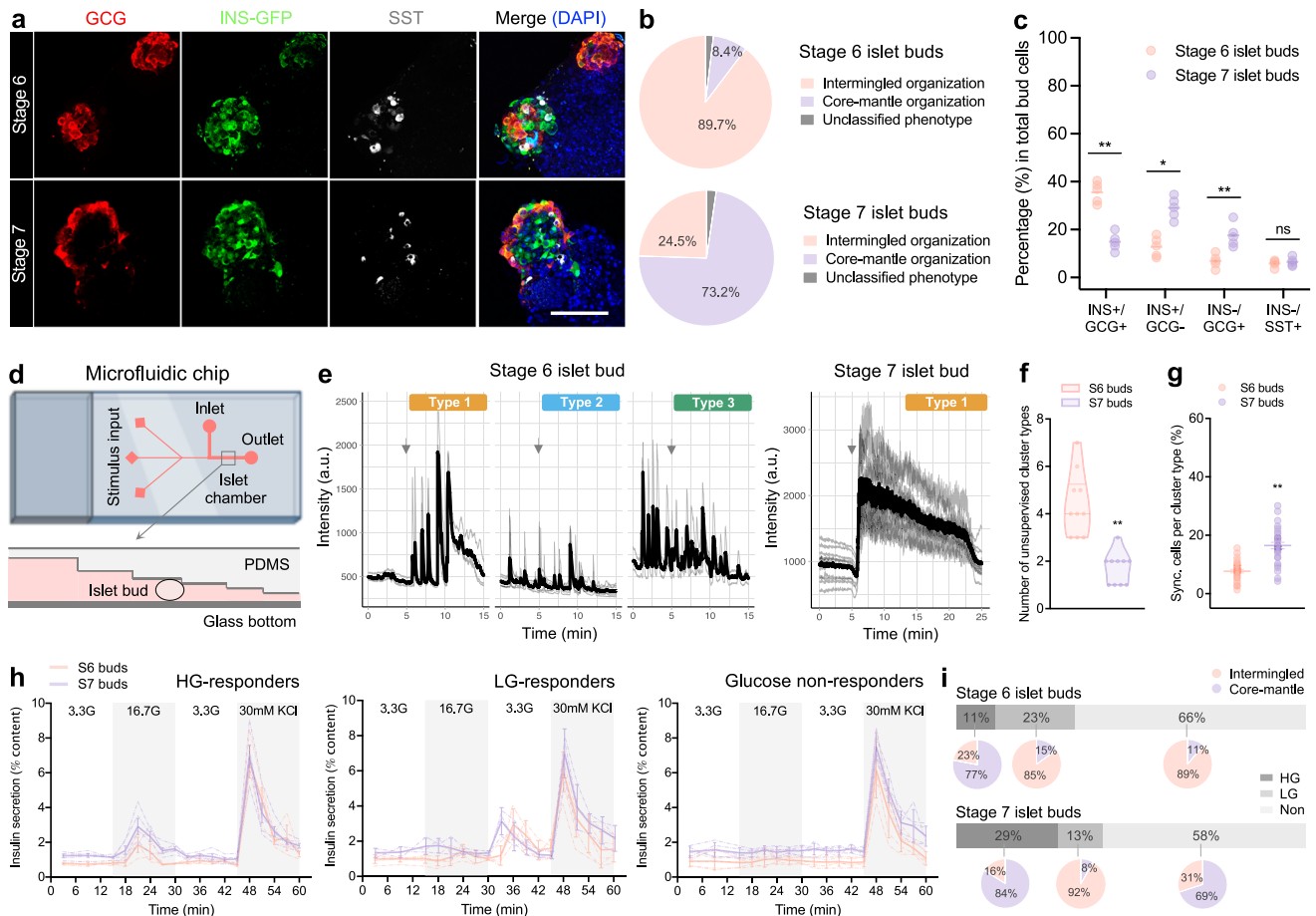

**Fig. 3 | Cytoarchitectural rearrangement and functional heterogeneity of differentiating islet buds. a** Representative images of typical Stage 6-7 clusters stained for major islet cell types. INS insulin, GCG glucagon, SST somatostatin. Nuclei were counterstained with DAPI (blue). Scale bar, 100 μm. **b** Quantification of the percentage of different cytoarchitectural types adopted by Stage 6-7 islet buds. $n = 30$–$50$ clusters for each from 5-8 independent differentiations. **c** Flow cytometry quantification showing the percentage of major islet cell composition in Stage 6-7 islet buds. $n = 5$ independent differentiations, ns not significant, *$p < 0.05$, **$p < 0.01$ versus Stage 6 islet buds, unpaired two-tailed t-test. **d** Schematic of microfluidic chip system (termed as "microperifusion") to perfuse single islet bud for imaging calcium activities and detecting dynamic insulin secretion. Created with BioRender.com released under a Creative Commons Attribution-NonCommercial-NoDerivs 4.0 International license. **e** Traces of glucose-stimulated calcium responses in representative Stage 6-7 islet buds were categorized by an

unsupervised clustering algorithm (see *Methods*). Glucose stimulation (16.7 mM) was added at the time point of 5-min (indicated by arrows). Quantification showing the number of cluster types (**f**) and the percentage of synchronized cells per cluster type (**g**) in Stage 6-7 islet buds. $n = 10$ clusters for each from 3 to 4 independent differentiations. **$p < 0.01$ versus Stage 6 islet buds, unpaired two-tailed t-test. **h** Microperifusion showing dynamic insulin secretion from Stage 6-7 islet buds, which were categorized to high glucose (HG)-responders, low glucose (LG)-responders, and glucose non-responders (Non). Traces were presented as mean value ± SEM ($n = 3$–$5$ clusters for each from 4 independent differentiations) and individual traces were also shown with dotted lines. **i** Quantification showing the percentages of glucose-responder types and architectural types adopted by Stage 6-7 buds. $n = 30$–$35$ clusters for each from 4 independent differentiations. Budding differentiation was induced by 100 ng/mL GDF8 plus 1–1.5 μM CHIR99021 for Stage 1 Day 1 using the Mel1 *INS*[GFP/W] line. Source data are provided as a Source Data file.

protocol (Fig. 4a–d). The reduction of PDX1+ cells to ~44% of *RFX6* KO Stage 4 cultures in the bulk pipeline predicted a switch to a budding-type differentiation pattern. Indeed, PDX1+ cell budding morphogenesis occurred (Fig. 4e). Despite the bud formation, subsequent commitment to CHGA+ endocrine precursors was markedly impaired (Fig. 4e, f). In the budding differentiation protocol, the number of PDX1+ cells was further decreased in *RFX6* KO Stage 4 cultures (16.3% ± 2.4%). Morphologically, these sporadic PDX1+ cells formed even smaller buds (Fig. 4e and Supplementary Fig. 11b). In both differentiation pipelines, cells with *RFX6* deficiency failed to secrete insulin or generate normal islet cell types, interestingly with the exception that PPY+ cells which seemed to develop normally. Rare INS+ and GCG+ cells were detected; however, they were INS+/GCG+ bi-hormonal cells, suggesting functional incompetency (Fig. 4e, g and Supplementary Fig. 11c, d). Taken together, our stem cell modeling not only corroborates previous findings but also uncovers an unexpected role of *RFX6* in regulating pancreatic cell patterning and potentially the early pancreas morphology.

## Distinct transcriptomic profiles of islet buds and main bodies

PDX1+ cell budding morphogenesis occurs in a heterogeneous cell population in our budding models. As islet cells were predominantly derived within the PDX1+ cell-enriched bud niche, we investigated the cellular identity of the main bodies (i.e., the PDX1-negative compartment) and the role of main body cells in islet bud differentiation. To address these questions, we purified islet buds and main bodies by enzymatic isolation and sorted the two compartments (PDXeG-positive buds and PDXeG-negative bodies) for bulk RNA sequencing (RNA-seq) (Fig. 5a, b). RNA-seq analysis revealed that bud and body cells exhibited very distinct transcriptomic profiles (Fig. 5c–e). Specifically, bud cells were enriched for *PDX1* and *NKX6.1* transcripts as well as expressed higher levels of genes associated with endocrine cell commitment (*NGN3, NEUROD1, CHGA, ISL1, RFX6, MNX1, FEV, PAX4, ARX*), islet cell types (*INS, GCG, GHRL*) and beta cell function (*GCK, G6PC2, ABCC8, KCNJ11, SIX2, UCN3*) (Fig. 5f). By contrast, main body cells expressed higher levels of genes associated with ductal cell markers (*SOX9, KRT18, MUC1, SPP1, TPM1*), cell proliferation (*PCNA, CDK2, MCM2, MCM3*), and

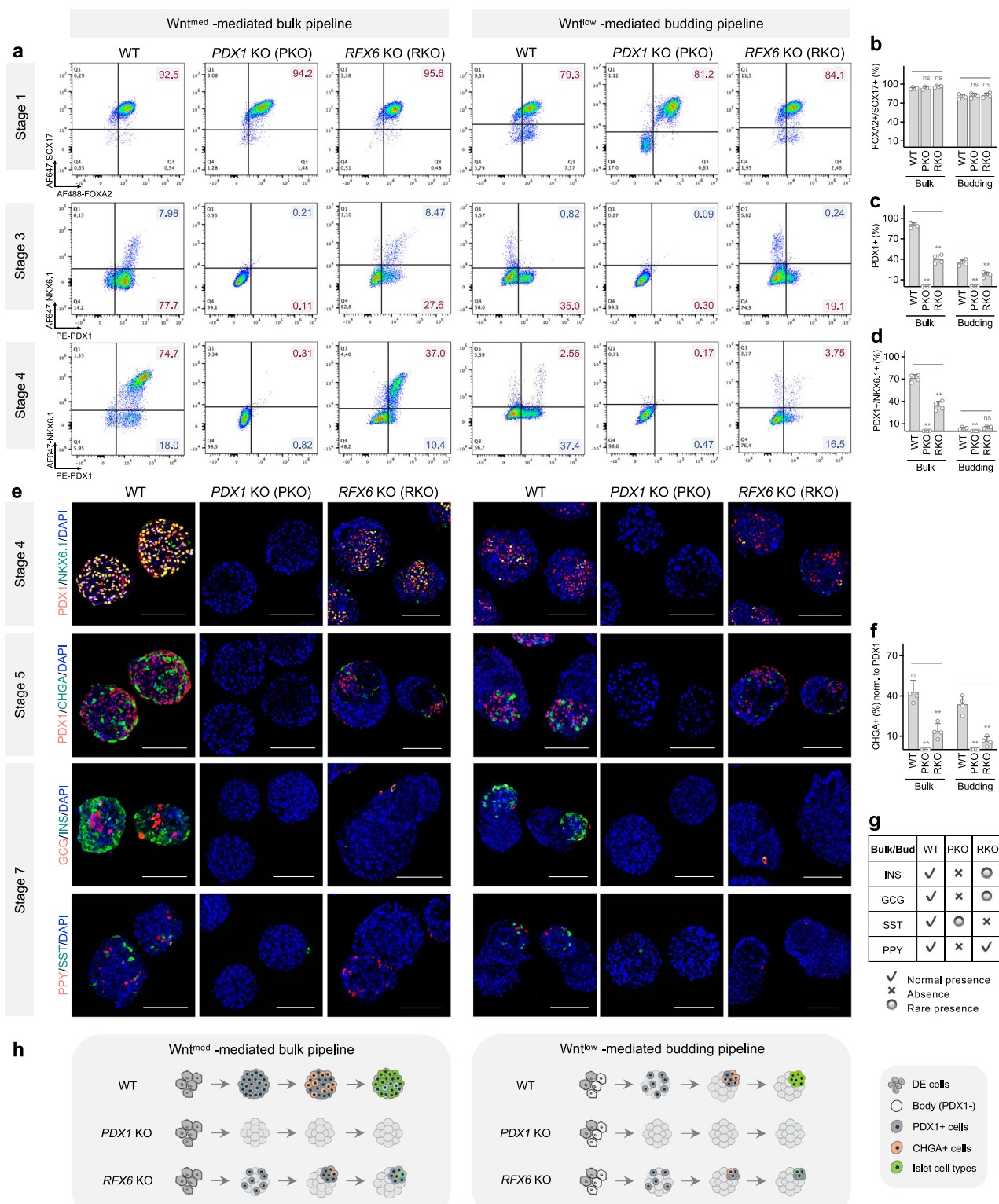

**Fig. 4 | The tunable differentiation system provides both transcriptional and morphological insights into modeling human pancreas diseases caused by *PDX1* and *RFX6* mutations.** Representative flow plots (**a**) and quantification of Stage 1 cells examined for FOXA2 and SOX17 (**b**), Stage 3-4 cells examined for PDX1 and NKX6.1 (**c**, **d**) from HUES8 wildtype (WT), *PDX1* KO and *RFX6* KO hESC-derived cultures. Both bulk and budding differentiation pipelines were applied. $n = 4$ independent differentiations, ns not significant, **$p < 0.01$, one-way ANOVA with Dunnett test for multiple comparisons to WT controls. **e** Representative images of typical Stage 4-7 clusters stained for pancreatic progenitor markers (PDX1, NKX6.1), endocrine marker (CHGA), and islet cell types (INS, GCG, SST, PPY) Nuclei were counterstained with DAPI (blue). Scale bars, 100 μm. **f** Quantification of CHGA+ cells in Stage 5 clusters. The percentage of CHGA+ cells was normalized to the total number of PDX1+ cells. $n = 4$ independent differentiations, ns not significant, **$p < 0.01$, one-way ANOVA with Dunnett test for multiple comparisons to WT controls. **g** Summary of the presence of islet cell types in WT, *PDX1* KO, and *RFX6* KO hESC-derived Stage 7 clusters. **h** Schematic summary of our stem cell modeling with WT, *PDX1* KO, and *RFX6* KO hESC lines. Created with BioRender.com released under a Creative Commons Attribution-NonCommercial-NoDerivs 4.0 International license. Bulk and budding differentiation were induced by 100 ng/mL GDF8 plus 3 μM and 1.5 μM CHIR99021 for Stage 1 Day 1, respectively, using the indicated hESC lines. Source data are provided as a Source Data file.

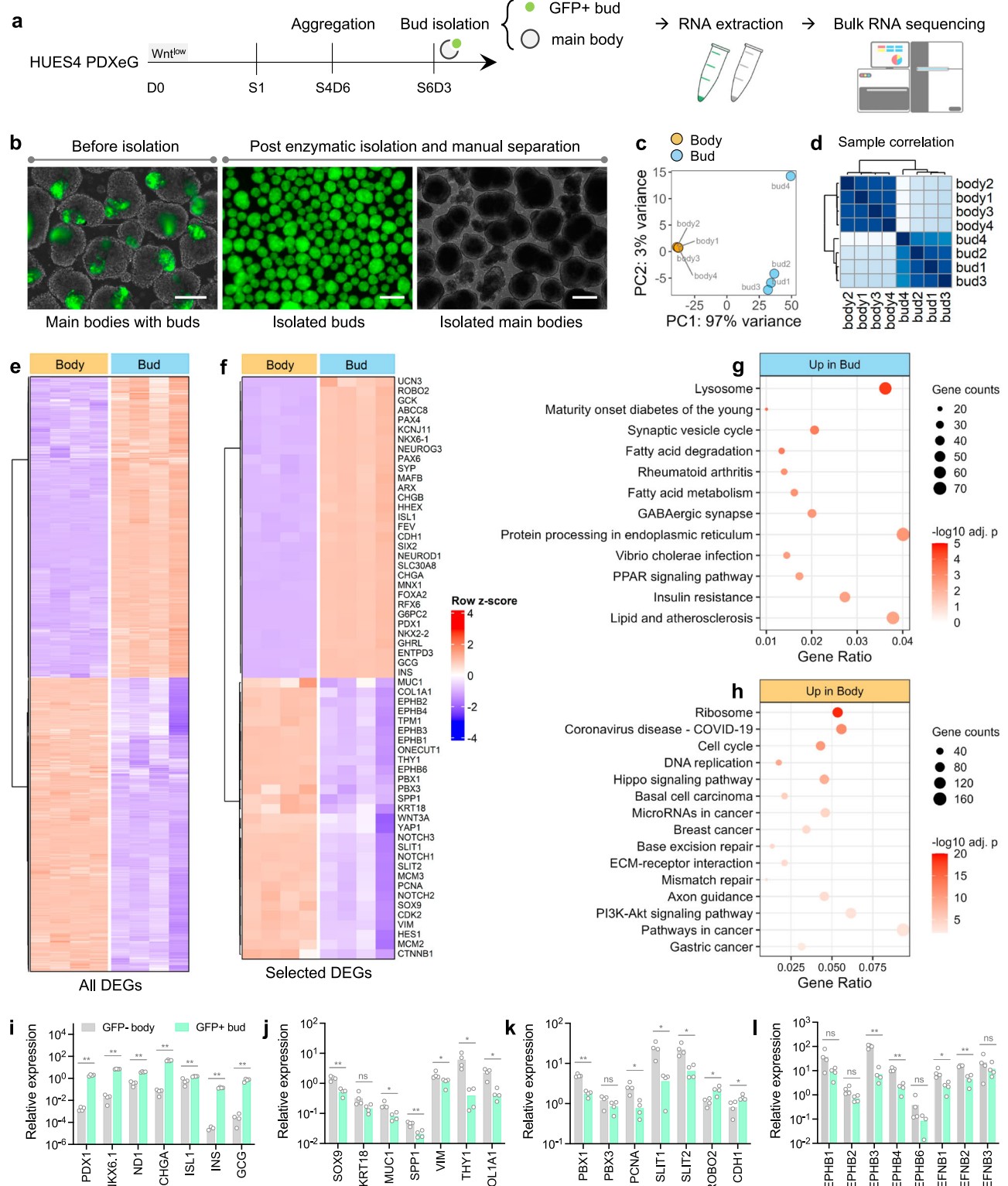

**Fig. 5 | RNA-seq reveals distinct transcriptomic profiles of islet buds and main bodies. a** Bulk RNA-seq design for characterizing islet bud and main body. Created with BioRender.com released under a Creative Commons Attribution-NonCommercial-NoDerivs 4.0 International license. **b** Representative images before and after enzymatic isolation (see Methods). The HUES4 PDXeG line was used for differentiation and for facilitating the sorting of GFP+ buds and GFP- bodies. Scale bars, 100 μm. Principal component analysis (**c**) and sample correlation (**d**) showing overall distinct transcriptomic profiles of islet buds and main bodies. $n = 4$ independent differentiations. Differential expression analysis showing all differentially expressed genes (DEGs) in (**e**), with selected DEGs in buds and main bodies highlighted in (**f**). Functional pathway enrichment analysis in buds (**g**) and main bodies (**h**). $n = 4$ independent batches of differentiations. Over-representation analysis with one-sided Fisher's exact test, $p$ values adjusted for multiple testing by Benjamini-Hochberg method. **i–l** qPCR assays validating expression of selected genes in buds and main bodies. $n = 4$ independent differentiations, ns not significant, $*p < 0.05$, $**p < 0.01$ versus GFP- body, unpaired two-tailed t-test. Budding differentiation was induced by 100 ng/mL GDF8 plus 1–1.5 μM CHIR99021 for Stage 1 Day 1 using the HUES4 PDXeG line. Source data are provided as a Source Data file.

pancreatic progenitor state maintenance (*YAP1, NOTCH1, NOTCH2, NOTCH3, HES1, WNT3A*) (Fig. 5f). Functional enrichment revealed that pathways of cell cycle, Hippo-YAP mechanosensitive signaling, extracellular matrix (ECM)-receptor interaction and axon guidance were up-regulated in main bodies (Fig. 5g, h). We confirmed the selected differentially expressed genes between buds and main bodies using qPCR (Fig. 5i–l).

In addition to the transcripts related to ductal lineage and pro-genitor state, we noted that main body cells expressed higher levels of pancreatic mesenchyme markers (*THY1, VIM, COL1A1, PBX1, PBX3*) as well as *SLIT1* and *SLIT2* (Fig. 5f, j, k). The Robo/Slit pathway is reported to be involved in many aspects of pancreas organogenesis and Slit ligands function as important pro-endocrine mesenchymal signal factors in mouse and human pancreatic tissues[38–40]. With *ROBO2* receptor robustly expressed in bud cells (Fig. 5f, k), the elevated expression of *SLIT1* and *SLIT2* in main body cells may provide surrounding cues beneficial to the differentiation and/or function of islet buds. However, Stage 6 islet buds separated from main bodies continued to differentiate and acquired comparable function (Supplementary Fig. 12), arguing against the requirement for crosstalk between the bud and main body during late stages, but an earlier role of body cells in bud morphogenesis/differentiation cannot be excluded.

To further characterize main body cells, we immunostained several differentially expressed candidates as informed by RNA-seq and qPCR analysis. Specifically, we revealed (1) remarkably higher expression levels of THY1 and VIM (pancreatic mesenchyme/fibroblast markers) in the main bodies; (2) mutually exclusive expression of the ductal cell marker SOX9 in main body cells and the endocrine cell marker NEUROD1 in bud cells; (3) higher expression level of YAP1 (a mechanosensitive signal for balancing progenitor cell self-renewal and differentiation) in the body cells versus lower expression of YAP1 in the NEUROD1+ endocrine-committed bud cells; and (4) higher expression of the anterior foregut marker SOX2 in body cells. We also examined the exocrine cell marker Trypsin 1/2/3 but did not detect any expression in either bud or body cells, ruling out the exocrine identity of main body cells (Supplementary Fig. 13). Collectively, these results suggest a potential pancreatic mesenchyme/fibroblast phenotype or anterior foregut/ductal lineage commitment but not exocrine lineage of the main body cells. Nevertheless, pathways associated with cell apoptosis, DNA base excision repair, and mismatch repair were highly enriched in main body cells (Fig. 5h), suggesting that the identity of these cells cannot be properly maintained when entering endocrine induction stages.

## Involvement of EphB3/4 signaling in the PDX1+ cell budding morphogenesis

Our RNA-seq results identified candidate signaling pathways that could be involved in the PDX1+ cell budding process. Indeed, we noted several pathways that are differentially enriched in the buds versus main bodies and are also known to mediate cell migration or cell compaction. These include TGFβ signaling (epithelial-to-mesenchymal transition, EMT)[41], RhoA/ROCK (cytoskeletal remodeling)[42], EphA/EphrinA[43–45], and Robo/Slit (axon guidance cues)[38]. However, addition of specific inhibitors or activators targeting these signaling pathways at the budding stage did not impact PDX1+ cell clustering (Supplementary Fig. 14). A recent study reports that Wnt signaling separates the progenitor and endocrine compartments[46]; however, modulation of the Wnt pathway by supplementing cell cultures with Wnt activators/inhibitors at the budding stage did not affect PDX1+ bud formation or bud/body segregation in our spheroid system (Supplementary Fig. 14).

Guided by RNA-seq data, we noted that among axon guidance cues, EphB/EphrinB were differentially expressed in buds and main bodies (Fig. 5f). qPCR and immunostaining confirmed higher expression levels of EphB (EPHB1, EPHB2, EPHB3, EPHB4, EPHB6) and EphrinB (EFNB1, EFNB2, EFNB3) in main bodies relative to islet buds (Fig. 5l and Supplementary Fig. 15). Notably, EphB/EphrinB is known to regulate cell sprouting, cell type segregation and boundary formation during tissue development[47–50]. For example, EphB3b/EphrinB1 signaling is reported to orient hepatoblasts migration and liver bud formation[51]. Particularly in the pancreas, EphB signaling is required for proper pancreatic epithelium branching morphogenesis and EphB3 is found to be transiently expressed in delaminating endocrine-committed cells[52,53]. To examine whether EphB signaling could affect PDX1+ cell clustering/budding morphogenesis, we added EphB inhibitors during Stage 5 when budding occurred (Fig. 6a). Strikingly, EphB3 and EphB4 specific inhibitors suppressed PDX1+ cell clustering in a dose-dependent manner, indicating that both EphB3 and EphB4 signaling are involved in the budding morphogenesis by preventing the inter-mixing of PDX1+ cells and non-endocrine main body cells (Supplementary Fig. 16). Indeed, PDX1+ cells displayed the highest extent of scattering when EphB signaling was blocked by pan-EphB inhibition (Supplementary Fig. 14). Compared to the clustered phenotype of PDX1+ cells in vehicle controls, surface plot analysis and live imaging showed interrupted PDX1+ cell clustering and the absence of budding morphogenesis when EphB3/4 signaling was perturbed (Fig. 6b–e).

Next, we assessed how disrupting PDX1+ cell budding morphogenesis impacts subsequent endocrine cell development in our stem cell differentiation systems. As the mechanism of action for EphB inhibitors is reversible, we transiently added EphB3/4 inhibitors during Stage 5 and removed inhibition at later stages (Fig. 6a). This EphB3/4 co-inhibition impaired islet bud formation and INS+ cell induction in Stage 6-7 cultures (Fig. 6f). Quantification showed that INS-GFP fluorescent intensities (mean values by Gaussian fitting: $36 \pm 2$ versus $50 \pm 3$ of control) and insulin secretion of Stage 7 islet buds that had been treated with EphB3/4 inhibitors were lower than those of vehicle control islet buds (Fig. 6g, h), indicating that the establishment of PDX1+ bud niche and endocrine aggregates is essential for beta cell formation and function. The transient inhibition of EphB signaling during Stage 5 did not impact the proportion of PDX1+ cells, but its effects on interrupting PDX1+ cell clustering were associated with reductions in pro-endocrine cells (NGN3+ cells: $6.7\% \pm 0.8\%$ in DMSO versus $1.7\% \pm 0.4\%$ in EphB3/4 co-inhibition; NEUROD1+ cells: $25.2\% \pm 1.1\%$ in DMSO versus $11.9\% \pm 0.7\%$ in EphB3/4 co-inhibition) and islet cell types (INS+ cells: $43\% \pm 2.6\%$ in DMSO versus $19.5\% \pm 3.7\%$ in EphB3/4 co-inhibition; GCG+ cells: $18.7\% \pm 1.2\%$ in DMSO versus $8.1\% \pm 1.6\%$ in EphB3/4 co-inhibition) (Fig. 6i–k). Altogether, these data demonstrate the involvement of EphB3/4 signaling in the PDX1+ cell budding morphogenesis and highlight the importance of the local niche in early pancreatic patterning as well as islet development.

## Discussion

While bulk-type differentiation of hPSCs is commonly used to generate islet clusters, we report here an islet budding-type differentiation system. We achieved this by a single-day treatment with fine-tuned doses of Wnt agonist at the endoderm stage, which when combined with our other optimized differentiation conditions, results in: (1) induction of PDX1+ cells in a heterogeneous hPSC-derived culture; (2) spontaneous clustering of PDX1+ cells to form a local pancreatic bud-like niche while segregating from PDX1 non-expressing compartment; (3) endocrine cell commitment and islet formation within the niche created by PDX1+ cell budding morphogenesis; and (4) cytoarchitectural rearrangement and functional heterogeneity of differentiating islet buds. Different from bulk-type differentiations, the budding system elicits a subset of hPSC-derived cells toward the endocrine lineage and creates a heterogeneous cell-cell interaction environment in the differentiating clusters. As such, this budding system provides a unique model to study asynchronous differentiation, complex interactions, and islet morphogenesis during pancreas development with human cells (Supplementary Table 1). This approach complements the

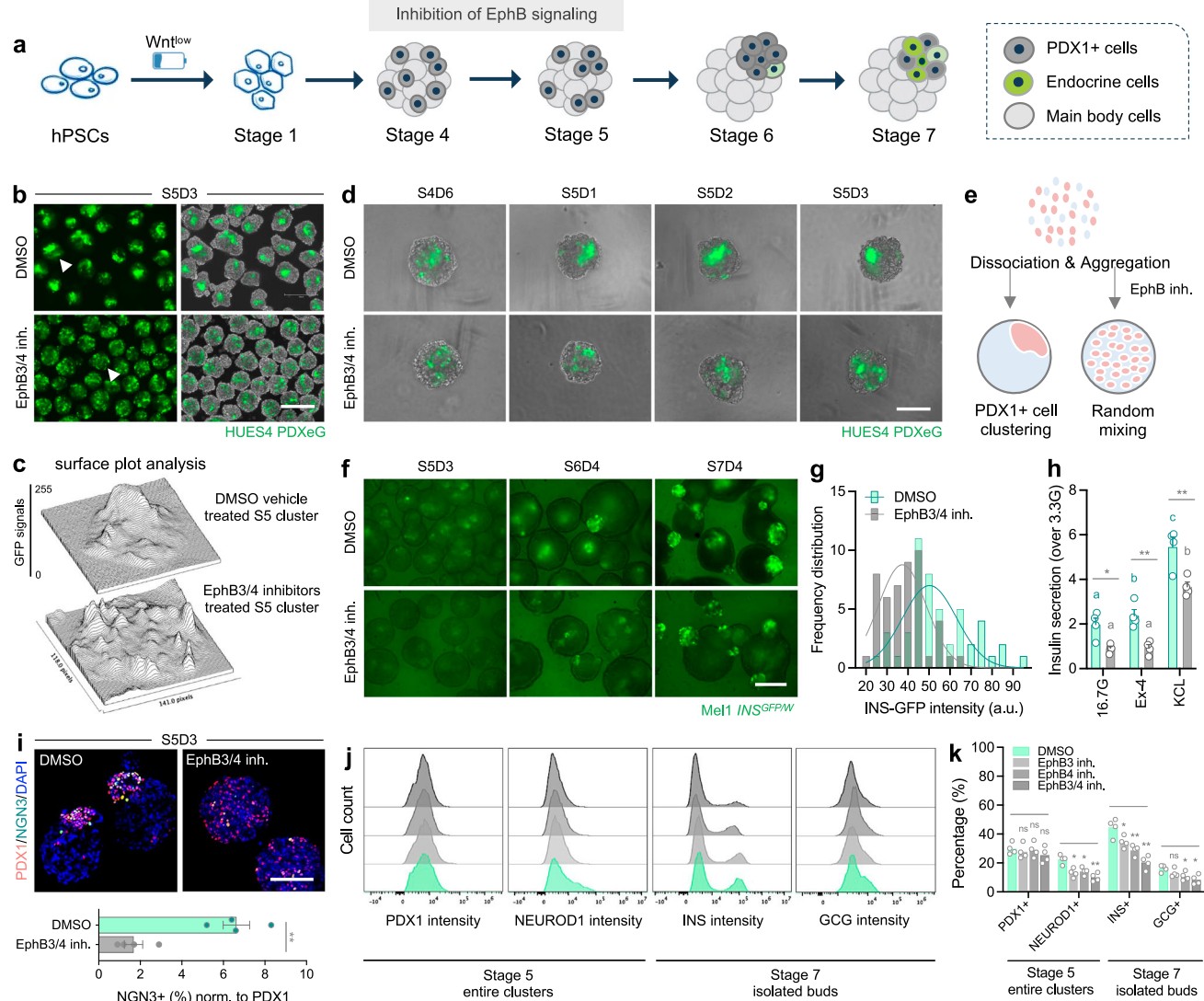

**Fig. 6 | Perturbation of EphB3/4 signaling disrupts PDX1+ cell clustering and compromises islet cell development. a** Study design for pharmacological perturbation of EphB signaling at budding stage. Created with BioRender.com released under a Creative Commons Attribution-NonCommercial-NoDerivs 4.0 International license. **b, c** Representative images (**b**) of Stage 5 clusters that were treated with DMSO vehicle or a combination of EphB3/4 inhibitors at budding stage. Surface plot analysis (**c**) showing PDX1+ cell distribution patterns in representative Stage 5 clusters as indicated by arrowheads in (**b**). Scale bar, 300 μm. EphB3/4 inhibition disrupted the consolidated structure of PDX1+ cells (**d**), and a schematic summary of phenotypes (**e**). Scale bar, 150 μm. Created with BioRender.com released under a Creative Commons Attribution-NonCommercial-NoDerivs 4.0 International license in (**e**). **f, g** Representative images (**f**) of Stage 5-7 clusters that were treated with DMSO or EphB3/4 inhibitors at budding stage. Quantification by histogram and Gaussian fitting (**g**) showing a left shift of INS-GFP intensities in Stage 7 clusters that had been treated with EphB3/4 inhibitors. $n = 50$ clusters for each from 4–5 independent differentiations. Scale bar, 150 μm. **h** Static GSIS assays showing insulin secretion from Stage 7 clusters that were treated with DMSO or

EphB3/4 inhibitors. $n = 4$ independent differentiations, unpaired two-tailed t-test (*$p < 0.05$, **$p < 0.01$) for comparisons between DMSO control and EphB3/4 inhibition at indicated secretagogue, one-way ANOVA with Tukey test (with different letters) for multiple comparisons between secretagogues within same treatment group. **i** Representative images of Stage 5 clusters (that were treated with DMSO or EphB3/4 inhibitors) stained for endocrine precursor marker NGN3. Data were normalized to PDX1. $n = 4$ independent differentiations, unpaired two-tailed t-test (**$p < 0.01$) for comparisons between DMSO control and EphB3/4 inhibition. Scale bar, 100 μm. Representative flow plots (**j**) and quantification (**k**) of Stage 5 entire clusters and Stage 7 isolated buds (that were treated with DMSO or EphB3/4 inhibitors) examined for various markers. $n = 4$ independent differentiations, ns not significant, *$p < 0.05$, **$p < 0.01$, one-way ANOVA with Dunnett test for multiple comparisons to DMSO control clusters. Budding differentiation was induced by 100 ng/mL GDF8 plus 1–1.5 μM CHIR99021 for Stage 1 Day 1 using the HUES4 PDXeG line (**b–d**) and Mel1 $INS^{GFP/W}$ line (**f–k**). Source data are provided as a Source Data file.

use of human fetal pancreas tissues and organoid models from human and rodent sources[54–56].

Although it is well recognized that early cell culture conditions impact the overall quality and consistency of subsequent differentiations, in this study we show how a single-day exposure to varied Wnt treatment at the DE stage can drastically affect downstream pancreatic cell patterning (Fig. 1a), indicating the dose-sensitive requirement for Wnt signaling in determining pancreatic lineage differentiation

patterns (in a bulk or budding type) in stem cell systems and cross-validating similar recent observations[57]. We identified that the Wnt dose used to initiate a budding differentiation is cell line dependent (Supplementary Fig. 4), likely because hPSCs have variable endogenous Wnt signaling[58], such that Wnt concentrations must be empirically determined on a case-by-case basis. The observation that stem cells within a culture behave differently in response to Wnt activation suggests a heterogeneous starting population of hPSCs and a

subpopulation with responsiveness to Wnt^low conditions; in other words, this subpopulation is Wnt-hypersensitive. It would be interesting to probe the molecular signatures of these Wnt-hypersensitive cells and understand why they are prone to pancreatic lineage commitment.

The Wnt^low condition during the DE stage leads to an induction of 30–40% PDX1+ progenitor cells. Interestingly, these PDX1+ cells spontaneously cluster together and bud out forming a local pancreatic bud-like niche in differentiation cultures. Live imaging of fish embryos has shown that convergence of PDX1+ cells along notochord is required for development of pancreatic dorsal cell mass, equivalent to dorsal bud of mammals[59,60]. Ectopic expression of *PDX1* selectively in the gut epithelium of chick embryos by electroporation causes cells to bud out from the gut, resembling pancreatic buds[61]. Consistent with these prior findings, we showed that PDX1+ cell budding morphogenesis can be established in a human stem cell model (Fig. 2). Previous studies report that controlled clustering of PDX1+ cells in micropatterned wells enhances expression of the transcription factor NKX6.1[42,62]. We thus reason that formation of PDX1+ cell-enriched bud niche may promote local endocrine cell development. Indeed, the buds sequentially expressed NKX6.1, NGN3, NEUROD1, CHGA, SYN and INS (Fig. 2f–h and Supplementary Fig. 3), in a progression similar to what we obtain in bulk-type differentiation process. To investigate the mechanism of PDX1+ cell budding, we performed RNA-seq on differentiating spheroids and identified candidate signaling pathways. Among them, EphB3/4 signaling stood out from our bioinformatic analysis with expression of all five Eph receptors and three Ephrin ligands verified in various assays (Fig. 5f, l and Supplementary Fig. 15, 17). Through pharmacological perturbation, we demonstrate the involvement of EphB3/4 signaling in the PDX1+ cell budding morphogenesis during early pancreatic patterning (Fig. 6 and Supplementary Fig. 14, 16). Nevertheless, the EphB3/4 co-inhibition treatment may not completely block EphB signaling, which could be a limitation of this approach. Although EphB signaling is known to prevent intermixing of different cell types and regulate boundary formation in many developing tissues[48–50], to our knowledge, this study suggests a previously unreported role of EphB3/4 signaling in pancreatic bud formation during islet development.

The observation that PDX1+ cells tend to have lower levels of Ephrin proteins (particularly EphrinB2 and EphrinB3) in budding-type Stage 5 clusters (Fig. 5 and Supplementary Fig. 15) may indicate an association between PDX1 and Ephrin expression. Interestingly, we find that all five Eph receptors and three Ephrin ligands are expressed in *PDX1* KO Stage 5 clusters (Supplementary Fig. 17). However, compared to the differential localization of Eph/Ephrin in endocrine buds and main bodies derived from wildtype cell line (Supplementary Fig. 15), we note the differential or polarized expression pattern is diminished in the *PDX1* KO clusters (Supplementary Fig. 17), suggesting that spatial expression pattern of Eph/Ephrin proteins is associated with the presence or absence of PDX1 and thus the budding morphology. It would be interesting to investigate as to whether PDX1 directly regulates Eph or Ephrin signaling and expression.

To gain some insights into attachment of the bud to the main body, we examined expression patterns of cell adhesion molecules, EMT, and cell polarity markers. We found strikingly elevated expression of adhesion molecules E-cadherin and beta-catenin in the endocrine buds (Supplementary Fig. 18), indicating an "epithelial" phenotype of bud cells and tight cell-cell contacts via adheren junctions within the bud compartment. COL4A1/A2, a major component of intra-islet basement membrane proteins, was only expressed in the buds (Supplementary Fig. 18), in line with its abundant presence in human islets. By contrast, higher expression levels of mesenchymal and fibroblast phenotype markers VIM and THY1 were seen in the main body compartment (Supplementary Fig. 13), again revealing distinct cellular identities of buds and main bodies. However, we did not detect expression of integrin (ITGA1), ZO-1 (a tight junction marker), or N-cadherin in the clusters, and none of these molecules examined were found to be enriched at the boundaries between buds and main bodies (Supplementary Fig. 18). Thus, it remains uncertain how the bud connects to the main body and requires further investigation.

The rearrangement of islet cells in differentiating Stage 6-7 islet spheroids correlated with heterogeneous functionality, as revealed by our microfluidic chip analysis of individual clusters (Fig. 3). Specifically, the fact that Stage 7 islet buds display better functionality than Stage 6 islet buds may emphasize that both architecture and cellular maturity states (e.g., mono- or bi-hormonal) determine islet function. The majority of Stage 6 islet cells were INS+/GCG+ bi-hormonal cells adopting an intermingled structure, whereas Stage 7 islet cells were mostly INS+/GCG- and INS-/GCG+ monohormonal cells with a favorable core-mantle organization (Fig. 3a–c). Moreover, the core-mantle architecture rendered beta cells with more homotypic cell-cell interactions, in association with more synchronized calcium activities under glucose stimulation (Fig. 3e–g), providing a plausible explanation why the core-mantle architecture performed better than a mixed one in our stem cell model. A previous study reported that small-sized human islets prefer to adopt a core-mantle pattern (similar to mouse islets) and large-sized islets with a mixed organization may form when "core-mantle modular units" coalesce, suggesting rearrangement of human islet cells[63]. Similarly, our 80-100 µm hPSC-derived islet buds often establish a beta cell-enriched core surrounded by a mantle of alpha cells. To explore possible contributing factors, we examined stage-specific additives and found that the combined use of R428, NAC and Trolox (unique recipes in Stage 7 medium, targeting Axl or antioxidants) appeared to induce the islet cell rearrangement to core-mantle structure, whereas omission or inclusion of Stage 6 unique components (LDN and GSiXX, inhibiting BMP and Notch signaling, respectively) did not (Supplementary Fig. 9). Strikingly, prolonged exposure to Stage 6 medium secured an intermingled organization of islet cells (Supplementary Fig. 9). Examination of cell composition in Stage 6-7 islet buds revealed a transition from INS+/GCG+ bi-hormonal cells to predominantly INS+ and GCG+ monohormonal cells over the course of differentiation (Fig. 3c), suggesting that cell maturation occurs during the process. Nevertheless, further investigation is required to address how this cytoarchitectural change occurs, for instance via cell migration and/or cell (trans-)differentiation, and how the structural change is linked with cell maturation. Such work could inform the observations of species differences in cytoarchitecture[64,65], rearrangement of islet cells in developing human fetal pancreas[66–68] and maintenance of cytoarchitecture in adult islets[38,69].

We assessed the utility of our islet spheroid systems in modeling monogenic human pancreas diseases. As proof-of-principle examples, we examined the outcomes following differentiation of two mutant hESC lines (PDX1 KO and RFX6 KO) using both bulk and budding differentiation pipelines (Fig. 4 and Supplementary Fig. 11). Not surprisingly based upon prior in vitro studies[32,70] and characterization of a human with an inactivating mutation in *PDX1*[31], *PDX1* KO hESCs failed to differentiate into pancreatic progenitors (and absence of budding morphogenesis) or endocrine cells, emphasizing the requirement for *PDX1* in early pancreatic patterning (Fig. 4h). When examining *RFX6* KO hESCs using our bulk differentiation protocol, we made the striking observation that *RFX6* KO Stage 4 cultures exhibited a PDX1+ cell budding morphogenesis akin to that of WT cells under Wnt^low conditions. However, these PDX1+ buds were restricted in further developing into most major islet cell types. Interestingly we found normal presence of PPY+ cells in *RFX6* KO hESC-derived Stage 7 clusters, consistent with the presence of these cells in the pancreas from *Rfx6* knockout mice[35]. In the budding differentiation scenario, *RFX6* deficiency resulted in an

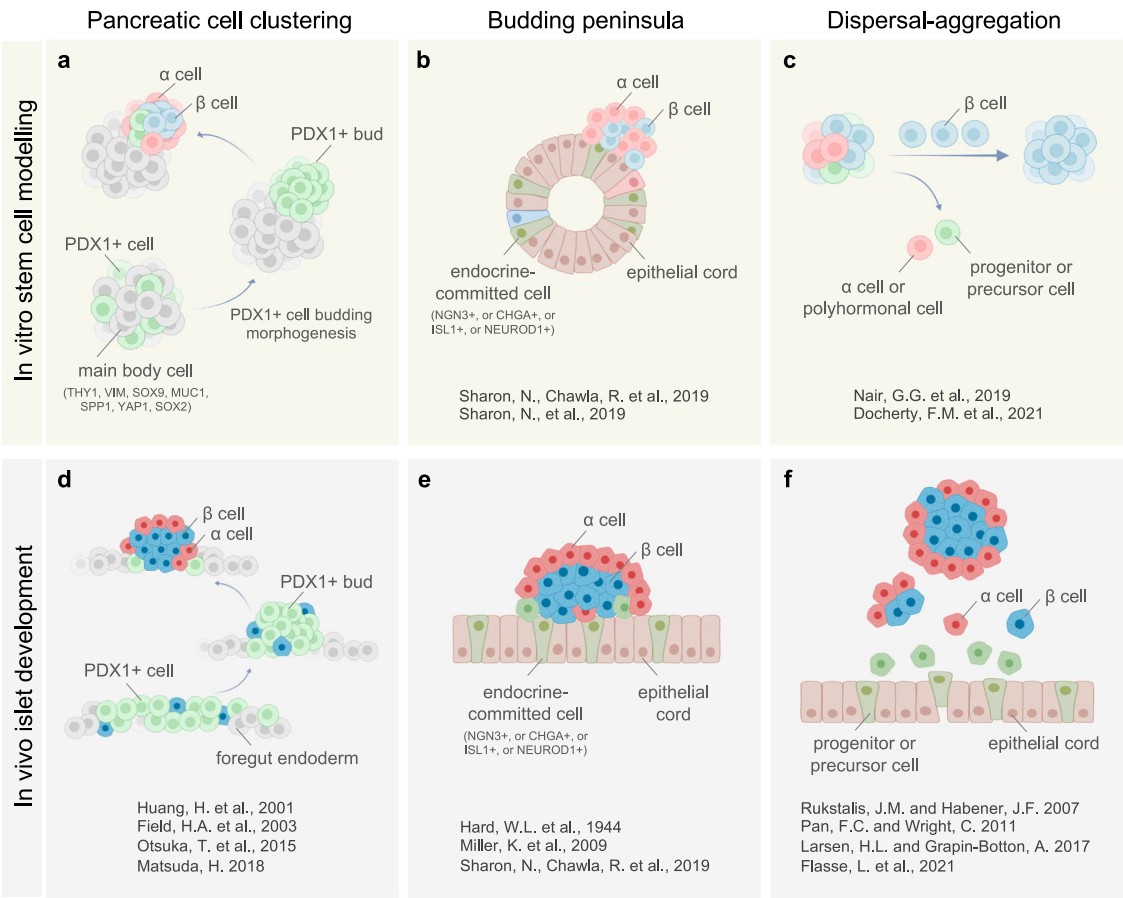

**Fig. 7 | A summary of three concepts for islet formation and morphogenesis.**
Schematic of three concepts for islet formation and morphogenesis, both in vitro
(**a**–**c**) and in vivo (**d**–**f**). The budding peninsula theory (**e**) proposes that islet
peninsulas form through coordinated migration and budding of endocrine-
committed cells, which emerge from a network of epithelial tubules, or "cords," and
remain attached to the outer layer. Therefore, peninsular growth relies on con-
tinuous recruitment of newly-formed endocrine-committed cells from epithelial
cords. In the dispersal-aggregation model (**f**), individual endocrine precursor cells
migrate away from epithelial cords via an EMT process and disperse into sur-
rounding mesenchyme. As cells differentiate and acquire an islet cell fate, they are
thought to aggregate into small clusters that later constitute complete islets. In our
differentiating stem cell model, we reveal that a vast majority of PDX1+ cells
undergo clustering and budding out from PDX1-negative main bodies prior to

endocrine specification. With our media highly enriched with endocrine cell-
directed cocktails (which inhibits progenitor cell proliferation and does not sup-
port ductal/exocrine commitment) at later stages, these hormone-negative, PDX1+
progenitor cells migrate together in an EphB3/4 signaling dependent manner, and
once aggregated, the PDX1+ pancreatic bud efficiently gives rise to an endocrine
islet while only a certain population of PDX1+ progenitor cells (very likely the ones
with high PDX1 expression level) are specified to islet cells within the bud (**a**). This
paradigm is supported by observations of islet formation in fish and lamprey,
where a dorsal bud is formed through the convergence of PDX1+ cells that later
develop into a principal islet (**d**). Created with BioRender.com released under a
Creative Commons Attribution-NonCommercial-NoDerivs 4.0 International license
in (**a**–**f**).

85% loss of PDX1+ cells and thus tiny buds formed with even fewer
scattered CHGA+ cells (Fig. 4 and Supplementary Fig. 11), repre-
senting a similar phenotype as seen in pancreatic histology of
autopsy samples from patients with Mitchell-Riley syndrome[34].

A recent study reports that *RFX6* haploinsufficiency impairs beta
cell function using an iPSC system harboring a specific *RFX6* loss-of-
function protein-truncating variant. We both show similar defects in
pancreatic progenitor specification (PDX1+/NKX6.1+ cells) and lack of
islet hormone production (INS+ and GCG+ cells), but PDX1+ cell bud-
ding morphogenesis or PPY+ cell commitment is not reported in their
model[71]. We speculate that different *RFX6* disruption systems and
culture formats used by the two studies may contribute to the differ-
ences in these independent findings. In sum, our stem cell modeling
corroborates the importance of *RFX6* in pancreatic lineage differ-
entiation, but also uniquely reveals how *RFX6* deficiency could impact
morphological changes in early PDX1+ cell patterning and its con-
sequences on islet morphogenesis. Collectively, these examples illus-
trate the uniqueness and reliability of our tunable stem cell systems for

modeling pancreas diseases and the power to uncover new
mechanisms.

Different concepts have been proposed to describe the process of
islet formation and morphogenesis both in vitro and in vivo (Fig. 7a–f).
In the prevailing notion of in vivo islet development, the "dispersal-
aggregation" (or endocrine cell clustering) mechanism[17–19,67,72], endo-
crine precursor cells migrate away from pancreatic epithelium via an
EMT process and appear scattered in surrounding mesenchyme. As
cells differentiate and acquire an islet cell fate, they are thought to
aggregate into small clusters that later constitute complete islets
(Fig. 7f). The recent "budding peninsula" theory supported by evi-
dence in mice[46,73] posits that islet peninsulas form through coordi-
nated migration and budding of endocrine-committed cells, which
emerge from a network of epithelial tubules, or "cords," and remain
attached to its outer surface[74,75]. Therefore, peninsula growth relies on
the continuous recruitment of recently formed endocrine-committed
cells from epithelial cords (Fig. 7e). Both principles have been used as
references to guide differentiation of hPSCs towards islet-like clusters

in vitro (Fig. 7b, c)[11,46]. Distinct from these two concepts, in our stem cell modeling we do not see islet formation through either dispersal-aggregation of individual endocrine cells or coordinated movement of endocrine-committed cells from distant places. Rather, with a focus on an earlier stage prior to endocrine specification, we reveal that hormone-negative, PDX1+ progenitor cells migrate together (termed "pancreatic cell clustering"), and once aggregated, a certain population of the progenitors subsequently develops into islet cells (Fig. 7a). This islet budding paradigm is supported by observations of islet formation in fish and lamprey models, where a dorsal bud is formed through convergence of PDX1+ cells that later develop into a single principal islet[59,60,76,77], suggesting that it could be a highly conserved biological mechanism.

In conclusion, we offer here an experimental model that captures multiple morphological events and provide additional insights into the processes of islet formation and morphogenesis. We demonstrate that: (1) in an asynchronous differentiation or multicellular environment, PDX1+ cells are able to come together and bud out (termed "PDX1+ cell budding morphogenesis"), a process that requires EphB3/4 signaling; (2) the PDX1+ cell-enriched budding structures create a local niche, resembling pancreatic buds, whereby PDX1+ progenitor cells are efficiently committed to pro-endocrine cells and islet cell types; (3) the spatial organization of islet cells can to some extent be regulated by morphogen signals (Axl, BMP and Notch) and antioxidants, and is tightly linked with functionality. There are limitations of this study. In our stem cell model, islets are induced to bud out from non-endocrine main bodies. Although the transcriptomic profiles of bud and body cells were characterized by RNA-seq, unlike bud cells, the exact cellular identity of main bodies and what they may contribute to the process of PDX1+ cell migration or any earlier role of them in islet bud morphogenesis/differentiation remains ambiguous. While our differentiation approach indicates that islets form in the local bud niche created by PDX1+ cell budding morphogenesis, the degree to which this budding process mimics human islet development is presently unclear and will need further investigation. Finally, this new experimental model may be used to address fundamental questions of islet biology in a human context, but its low yield makes it unsuitable for application as cell replacement therapy.

## Methods

### Cell culture and tunable differentiation of hPSCs

The H1 hESC line was obtained from WiCell[78]; Mel1 $INS^{GFP/W}$ line was provided by Dr. Edouard G. Stanley from MCRI and Monash University; HUES4 PDXeG line was provided by Dr. Henrik Semb; GCaMP_CRISPRi hiPSC line was provided by Dr. Knut Woltjen and Dr. Bruce Conklin[79]. HUES8 iCas9 parental, *PDX1* KO, and *RFX6* KO lines were provided by Dr. Danwei Huangfu. The reliability of Mel1 $INS^{GFP/W}$ and HUES4 PDXeG reporter lines was validated by both immunostaining and flow cytometry (Supplementary Fig. 19). Human islets were obtained from the Alberta Diabetes Institute (ADI) Islet Core with informed consent and cultured in CIT medium (Corning) as previously described[23]. Refer to Supplementary Table 2 and *Reporting Summary* for cell line and human islet details. All work involving hPSCs and human islets carried out in this study were approved by the University of British Columbia Clinical Research Ethics Board and the Canadian Stem Cell Oversight Committee with appropriate consent and conditions.

Undifferentiated hPSCs tested negative for mycoplasma were maintained on hESC-qualified Matrigel (Corning) and cultured in mTeSR1 complete medium (STEMCELL Technologies) as previously described[23,80]. Of note, Matrigel coating was only used for the maintenance culture and planar culture during the first four stages of differentiation. Bud induction and subsequent endocrine cell differentiation during the last three stages were conducted under static suspension culture without Matrigel coating or mounting. Specifically, differentiation was initiated 24 h after seeding with a density

of $1.5–2 \times 10^5$ cells per $cm^2$ depending on cell lines when hPSC cultures reached 95–100% starting confluence. For Stage 1, hPSC cultures were exposed to different Wnt conditions for inducing either bulk- or budding-type differentiation. Specifically, to initiate bulk-type differentiation, cells were exposed to Basal Medium 1 supplemented with 100 ng/mL GDF-8 (PeproTech) and 3 µM CHIR99021 (GSK3β inhibitor, Sigma) for day 1 only; for day 2–3, cells were cultured in Basal Medium 1 supplemented with 100 ng/mL GDF-8. This Wnt$^{med}$ condition during Stage 1 resulted in robust induction of DE cells and was used as the lowest concentration to induce bulk-type differentiation (i.e., PDX1+ buds are no longer formed under this condition). To initiate budding-type differentiation, cells were exposed to Basal Medium 1 supplemented with 100 ng/mL GDF-8 and 0.2–1.5 µM CHIR99021 (depending on cell lines) for day 1 only; for day 2–3, cells were cultured in Basal Medium 1 supplemented with 100 ng/mL GDF-8. This Wnt$^{low}$ condition during Stage 1 is a key step for inducing budding-type differentiation. Other cocktails and Wnt agonist concentrations used at Stage 1 Day 1 for tuning to bulk- or budding-type differentiation were specified in Supplementary Fig. 4a. For remaining Stages 2-7, bulk and budding differentiation followed exactly the same protocol. Briefly, Stage 1 cells from bulk or budding cultures were sequentially exposed to Stage 2 medium for 2 days, Stage 3 medium for 2 days, and Stage 4 medium for 5–6 days. On the last day of Stage 4, cultures were either manually scaped into small clumps as previously described or completely dissociated with Accutase (STEMCELL Technologies) for 12–15 min at 37 °C, rinsed with DMEM/F-12 (Thermo Fisher Scientific) and spun at 300 rcf for 5 min. For the Accutase method, the resulting single cell pellet was resuspended in Stage 4 medium plus 10 µM Y-27632 (STEMCELL Technologies) and aggregated using AggreWell™–400/800 microwell plates (STEMCELL Technologies) following the manufacturer's protocol. Each microwell was seeded with 2000-3000 cells and aggregated for 24 h in a humidified incubator at 5% $CO_2$ at 37 °C for cluster formation. The resulting Stage 4 clusters were retrieved from microwells and transferred into ultralow attachment (ULA) flat bottom 96-wells for static suspension culture during Stages 5-7 as we recently reported[23,24,80]. Briefly, ~600 Stage 4 clusters from two wells of AggreWell™–800 plate were resuspended in 3 mL Stage 5 medium resulting in a density of 20 clusters per 100 µL. Fifty microliters Stage 5 medium was added into each well of ULA 96-well plate and then 100 µL of the cluster resuspension (containing ~20 clusters) was transferred into each well and sparsely distributed, resulting in 150 µL culture medium in total per well. Culture plates were placed on a level surface in a humidified incubator at 5% $CO_2$ at 37 °C. Clusters were sequentially exposed to Stage 5 medium for 3 days, Stage 6 medium for 6–8 days, and Stage 7 medium for 8–10 days. Two-thirds of the medium (~100 µL) was changed daily. In some cases, we handpicked individual clusters and transferred them to an ULA 96-well plate (1 cluster per well) for cell/cluster tracking analysis.

Differentiation media used for each stage was as follows. Basal Medium 1: MCDB131 medium (Life Technologies) supplemented with 1.5 g/L NaHCO₃ (Sigma), 1× Glutamax (Thermo Fisher Scientific), 0.5% fatty acid-free BSA (Proliant) and 5 mM glucose (Sigma). Basal Medium 2: MCDB131 medium supplemented with 2.5 g/L NaHCO₃, 1× Glutamax, 0.5× ITS-X (Thermo Fisher Scientific), 2% fatty acid-free BSA and 5 mM glucose. Basal Medium 3: MCDB131 medium supplemented with 1.5 g/L NaHCO₃, 1× Glutamax, 0.5× ITS-X, 2% fatty acid-free BSA and 15 mM glucose. Stage 1 medium: formulations were detailed above for bulk and budding protocols. Stage 2 medium: Basal Medium 1 supplemented with 0.25 mM L-ascorbic acid (Sigma) and 50 ng/mL FGF7 (R&D Systems). Stage 3 medium: Basal Medium 2 supplemented with 0.25 mM L-ascorbic acid, 50 ng/mL FGF7, 0.25 µM SANT-1 (Sigma), 1 µM retinoic acid (Sigma), 100 nM LDN193189 (STEMCELL Technologies) and 200 nM TPPB (Tocris). Stage 4 medium: Basal Medium 2 supplemented with 0.25 mM L-ascorbic acid, 2 ng/mL FGF7, 0.25 µM SANT-1, 0.1 µM retinoic acid, 200 nM LDN193189 and 100 nM TPPB.

Stage 5 medium: Basal Medium 3 supplemented with 0.25 µM SANT-1, 0.05 µM retinoic acid, 100 nM LDN193189, 1 µM T3 (Sigma), 10 µM ALK5 inhibitor II (Cayman Chemicals), 10 µM zinc sulfate (Sigma) and 10 µg/mL of heparin (Sigma). Stage 6 medium: Basal Medium 3 supplemented with 100 nM LDN193189, 1 µM T3, 10 µM ALK5 inhibitor II, 100 nM GSi XX (Sigma), 10 µM zinc sulfate and 10 µg/mL of heparin. Stage 7 medium: Basal Medium 3 supplemented with 1 µM T3, 10 µM ALK5 inhibitor II, 1 mM NAC (Sigma), 10 µM Trolox (Sigma), 2 µM R428 (AXL inhibitor, Cayman Chemicals), 10 µM zinc sulfate and 10 µg/mL of heparin. Refer to Supplementary Table 2 for details.

### Enzymatic isolation of intact islet buds from main bodies

From technical aspect, this enzymatic method is only effective to isolate intact islet from main bodies after early Stage 6, when the buds become more protruding from main bodies. Specifically, to separate islet buds and main bodies, the Stage 6/7 spheroids were rinsed with HBSS (without $Ca^{2+}$ and $Mg^{2+}$, Sigma) and incubated with freshly-made collagenase P solution (1 mg/mL in HBSS) at 37 °C water bath for 12–15 min. After gentle pipetting up and down 20–30 times, digestion was stopped by adding equal volume of ice-cold HBSS (containing 20% FBS). Buds and main bodies were handpicked and manually sorted under a fluorescent microscope in biosafety cabinet. Isolated buds and main bodies were further differentiated, or used for downstream assays such as bulk RNA sequencing and microperifusion.

### Eph inhibition for disrupting PDX1+ cell clustering

Clusters were treated with different Eph inhibitors for 3 days during Stage 5, at which time point PDX1+ cell clustering occurred in the budding-type differentiation cultures. Dose responses of each Eph inhibitor were carefully examined. Specifically, pan-Eph RTK inhibitor MG-516 (a potent inhibitor for blocking Eph phosphorylation) and pan-EphA inhibitor (pyrrolyl benzoic acid, PBA) were used for initial perturbation tests. As PBA treatment did not impact PDX1+ cell clustering in the initial tests, we concluded that EphA signaling was not required for this process. Thus, selective EphA inhibition was not further investigated. For selective EphB inhibition experiments, 0.1–10 µM LDN-211904 (Millipore) were used to inhibit EphB3 signaling and 0.1–1 µM NVP-BHG712 (SelleckChem) were used to inhibit EphB4 signaling. Co-inhibition of EphB3/4 signaling was done with a combination of 1 µM LDN-211904 and 1 µM NVP-BHG712.

### Flow cytometry

Stages 1 and 4-7 cultures were dissociated using Accutase and incubated for 10–12 min at 37 °C to generate single cell suspension. After washing twice with FACS buffer (PBS plus 2% FBS), samples were fixed with BD Biosciences Fix/Perm buffer (BD Biosciences) for 10 min at room temperature followed by two washes in 1× Perm/Wash buffer (made by diluting 10× BD Biosciences Perm/Wash buffer into deionized water). Cells were then stained for intracellular markers using various stains diluted in 1× BD Biosciences Perm/Wash buffer for 45 min at room temperature and kept in the dark. Each antibody was carefully titrated with stem cell controls, and gating strategy was determined using unstained and/or isotype controls (an example of gating strategy is shown in Supplementary Fig. 20). Flow cytometric analysis was performed on the CytoFLEX flow cytometer (Beckman Coulter) and FlowJo v. 10.1 software (FlowJo, LLC) was used for data analysis. Antibodies and dilutions are listed on Supplementary Table 2.

### Quantitative PCR

QIAzol reagent (Qiagen) or RNeasy® Mini Kit (Qiagen) was used to extract total RNA according to the manufacturer's protocol. iScript™ gDNA Clear cDNA Synthesis kit (Bio-Rad) was used to generate cDNA. Primer pairs were ordered from the Integrated DNA Technologies Inc. qPCR reactions were performed using SsoFast™ EvaGreen® Supermix (Bio-Rad) on the Bio-Rad CFX96 Real-Time System. For cycling conditions, amplification was done by 95 °C for 1 min followed by 40 cycles of 95 °C for 5 s plus 60 °C for 25 s. The melting curve was done by ramping from 65 °C to 95 °C with increasing by 0.5 °C/s and measuring relative fluorescence units at each 0.5 °C increment. Data were normalized to the housekeeping gene NFX1, and relative gene expression was presented by comparison to the expression level of human islet controls. TaqMan® and GeneQuery® Human Wnt Signaling Pathway qPCR Array 96-Well Plates were used to examine expression of WNT pathway genes according to the manual guide. Graphs were generated using Heatmapper and NetworkAnalyst[81,82]. Primer sequences are listed on Supplementary Table 3.

### Immunostaining

For whole mount staining, clusters were fixed overnight in 4% PFA at 4 °C. Clusters were rinsed with PBST (0.1% Triton-X in PBS) and permeabilized in 0.3% Triton-X in PBS overnight at room temperature (RT). Clusters were then blocked for 1 h with 5% BSA in 0.3% Triton-X and stained with primary antibodies diluted in 1% BSA in 0.3% Triton-X overnight at RT. Clusters were washed three times (each for 30 min) with PBST followed by secondary antibody staining in 1% BSA in 0.3% Triton-X overnight at RT. After a brief rinse with PBST, nuclei were stained with 10 µg/mL DAPI (Sigma) in PBST for 15 min and then washed three times with PBST at 30 min intervals. A tilt shaker was used for gentle agitation during incubation and washing steps following fixation. After the staining procedure, clusters were attached onto confocal chamber slides using Cell-Tak tissue adhesive (Corning) and then mounted with tissue clearing solution (80% glycerol in PBS) and covered with coverslips. Confocal laser scanning imaging was performed with a Leica TCS SP5 microscope with Leica Application Suite AF using PMT detectors and a HCX PL APO CS 40.0/1.25 NA OIL UV objective with lasers 405 nm, 488 nm, 561 nm and 633 nm. Image processing and analysis were performed using Fiji software as previously described[23,80,83–85]. Antibodies and dilutions are listed on Supplementary Table 2. Antibody validation is specified in the *Reporting Summary* file.

For paraffin section staining, fixed clusters were embedded in 1% agarose and stored in 70% ethanol. Samples were then embedded in paraffin and sectioned at 5 µm thickness (Wax-it Histology Services). Slides were deparaffinized and hydrated with xylenes followed by a graded alcohol series. Sections were subjected to 15-min heat-induced epitope retrieval at 95 °C in 10 mM citrate buffer plus 0.05% Tween-20 (pH 6.0). Following a 10-min blocking step with Protein Block (DAKO), slides were stained overnight at 4 °C with primary antibody diluted in Antibody Diluent (DAKO). Slides were washed three times with PBS and stained for 1 h with diluted secondary antibodies at room temperature. After staining, slides were washed three times with PBS and mounted using VECTASHIELD® HardSet™ Antifade Mounting Medium with DAPI (Vector Laboratories). Whole slide scanning was performed using the ImageXpressMicro™ Imaging System (Molecular Devices Corporation). Images were stitched and analyzed using the MetaXpress Software.

### Dithizone staining

hPSC-islet buds were washed once with DPBS and then stained with 5 mg/mL dithizone solution (Sigma-Aldrich) for 2–3 min at room temperature. After staining, clusters were washed extensively with DPBS and imaged under Zeiss Axio Zoom.V16 microscope (Zeiss).

### Static glucose-stimulated insulin secretion (GSIS) assay

Static GSIS assay of hPSC-islet spheroids was performed according to our previously described protocol[23]. Briefly, clusters were equilibrated in low glucose KRB buffer (129 mM NaCl, 4.7 mM KCl, 2.5 mM $CaCl_2$, 1.2 mM $MgSO_4$, 1.2 mM $KH_2PO_4$, 5 mM $NaHCO_3$, 3.3 mM glucose, 10 mM HEPES and 0.5% BSA in deionized water, pH 7.4) for 1 h at 37 °C in a 5% $CO_2$ incubator. After equilibration, five clusters were incubated

in 100 μL low glucose KRB buffer for 30 min at 37 °C in a 5% $CO_2$ incubator. Supernatants (-100 μL) were collected and stored at −30 °C until measurement as basal insulin secretion. The same clusters were then incubated in 100 μL high glucose KRB buffer (129 mM NaCl, 4.7 mM KCl, 2.5 mM $CaCl_2$, 1.2 mM $MgSO_4$, 1.2 mM $KH_2PO_4$, 5 mM $NaHCO_3$, 16.7 mM glucose, 10 mM HEPES and 0.5% BSA in deionized water, pH 7.4) for 30 min at 37 °C in a 5% $CO_2$ incubator. Supernatants (-100 μL) were collected and stored at −30 °C until measurement as the glucose stimulated insulin secretion. The same clusters were then incubated in 100 μL high glucose KRB buffer plus 10 nM exendin-4 for 30 min at 37 °C in a 5% $CO_2$ incubator. Supernatants (-100 μL) were collected and stored at −30 °C until measurement as the glucose and incretin co-stimulated insulin secretion. The same clusters were lastly incubated in 100 μL 30 mM KCl KRB buffer (103.7 mM NaCl, 30 mM KCl, 2.5 mM $CaCl_2$, 1.2 mM $MgSO_4$, 1.2 mM $KH_2PO_4$, 5 mM $NaHCO_3$, 3.3 mM glucose, 10 mM HEPES and 0.5% BSA in deionized water, pH 7.4) for 30 min at 37 °C in a 5% $CO_2$ incubator, after which -100 μL supernatants were collected and stored at −30 °C until measurement as the KCl stimulated insulin secretion. For data normalization, clusters were lysed with RIPA lysis buffer for total insulin content extraction at the end of the assay. Human insulin in supernatant samples was measured using Human Insulin ELISA Kit (ALPCO) according to the manufacturer's protocol.

### Microfluidic chip perifusion
Microfluidic chips were kindly provided by Dr. Chao Tang and manufactured as previously described[28]. Before experiments, chips were degassed using a vacuum pump for 5 min and each channel connected with pipe was pre-filled with solutions. For in-chip calcium imaging, hPSC-islet buds were stained with 5 μM Cal-520 AM dye (in low glucose KRB buffer) for 45 min followed by brief rinse and another 15 min incubation in low glucose KRB buffer without the dye at 37 °C in a 5% $CO_2$ incubator. After staining, individual hPSC-islet buds were loaded into the microfluidic chip using a P10 pipette. The chip was then placed on the stage of Zeiss 3i spinning-disc confocal microscope with temperature control and perfused with low glucose KRB buffer for 1 h as equilibration and setup stabilization. Reagents were automatically pumped into the chip with a flow rate of 10 μL/min by program-controlled syringe pump (Harvard Apparatus). During imaging, the in-chip islet was kept at 37 °C all the time and sequentially perfused with low glucose KRB buffer for 5 min and high glucose KRB buffer for 10–20 min. A laser with 488 nm wavelength was used to illuminate Cal-520 signals and emission was collected using a single band pass filter (525 nm wavelength, 50 nm band width). Timelapse images were captured every 0.6 s with 100 ms exposure time with a Quant-EM 512SC Photometrics EMCCD camera under 40x/1.25 NA objective using SlideBook 6 software. Image analysis was performed using Fiji software[86] and unsupervised clustering of calcium traces was done using the online PlotTwist[29].

For in-chip perifusion for insulin secretion measurement, the islet-containing chip was placed on a heat block set at 37 °C and perfused with low glucose KRB buffer for 30 min as equilibration. Reagents were automatically pumped into the chip with a flow rate of 10 μL/min by program-controlled syringe pump. During perifusion, the in-chip islet buds were kept at 37 °C and sequentially perfused with low glucose KRB buffer for 15 min, high glucose KRB buffer for 15 min, low glucose for 15 min, and 30 mM KCl KRB buffer. Perfusates were manually collected every 3 min into collection plates and stored at −30 °C until assay. After perifusion, islet buds were retrieved for either hormone extraction or whole mount immunostaining. Human insulin was measured using Human Insulin ELISA Kit (ALPCO) following the manufacturer's protocol.

### Bulk RNA sequencing and data analysis
Total RNA from isolated buds and bodies was extracted and purified using QIAzol reagent according to the manufacturer's protocol.

Quality control (QC) test of RNA samples was performed using Agilent 2100 Bioanalyzer (Agilent). Samples passing the QC test (RIN > 8.0) were then library prepared following the standard protocol for NEB-Next Ultra II Stranded mRNA (New England Biolabs). Sequencing was performed on Illumina NextSeq500 sequencer with 42 bp × 42 bp paired-end reads and completed at BRC Sequencing Core facility at the University of British Columbia. Sequencing data were de-multiplexed using Illumina's bcl2fastq2 and the de-multiplexed reads were then aligned to the Homo sapiens (UCSC hg19) reference genome using STAR (Aligner) software (STAR_2.5.0b version). Assembly was done using Cufflinks software (2.2.1 version) with default settings on the Illumina Sequence Hub.

RNA-seq results were analyzed by DESeq2 R package[87], including the PCA, sample distance matrix, and differential expression. Genes that were detected with ≥5 raw counts in ≥25% samples were kept, and raw gene counts were normalized by variance stabilizing transformation. Normalized counts were used for PCA and sample distance matrix. For differential expression analysis, bud samples were contrasted against body samples by default DESeq2 method. P values were adjusted by Benjamini-Hochberg procedure, and genes with adjusted $p < 0.05$ were deemed to be differentially expressed. All the differentially expressed genes were plotted on a heatmap using ComplexHeatmap R package[88], with manually selected genes-of-interest labeled. For functional pathway enrichment analysis, KEGG pathways were enriched (over-represented) from genes that were significantly upregulated in the buds or upregulated in the bodies (downregulated in the buds), using clusterProfiler R package[89]. All the detected genes that passed the low count filter (≥5 raw counts in ≥25% samples) were used as the background gene list.

### Quantification and statistical analysis
Statistical analyses were performed using GraphPad Prism 10 software. Unpaired two-tailed t-test was used to compare differences between two indicated groups. For three or more groups, one-way ANOVA with Dunnett test was used to compare each group to a common control, or Tukey test was used to compare between groups. Sample size (n) indicates the number of independent biological replicates. Data are presented as mean value ± SEM. The following convention is used for indicating $p$ values and considered statistically significant: *$p < 0.05$, **$p < 0.01$. For all experiments, at least three independent biological replicates were conducted with similar results to confirm phenotypes and to ensure the reproducibility of our findings. See also *Reporting Summary* file.

### Reporting summary
Further information on research design is available in the Nature Portfolio Reporting Summary linked to this article.

## Data availability
RNA sequencing data generated in this study was deposited to the NCBI Gene Expression Omnibus (GEO) under the publicly available accession number GSE249020. Source data are provided with this paper. Refer to the Source Data file for all data generated and analyzed in this published article.

## Code availability
Code for data analysis was deposited to the GitHub (https://github.com/hcen/RNAseq_budding) with the https://doi.org/10.5281/zenodo.11201013, which includes all information required to reanalyze the data.

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

## Acknowledgements

This work was generously supported by funding from Canadian Institutes of Health Research, JDRF, Diabetes Canada, and STEMCELL Technologies. J.Z. is recipient of a NSERC-CREATE-JDRF Trainee Award. J.Z., S.L., and H.H.C. are recipients of Michael Smith Health Research BC Trainee Awards. J.Z. received travel awards for this work from the Stem Cell Network and BC Diabetes Research Network. We thank Kieffer lab members for many helpful discussions. We thank all collaborators and peers who generously shared experimental materials with us. Human islets for research were provided by the Alberta Diabetes Institute IsletCore with the support of the Human Organ Procurement and Exchange program, Trillium Gift of Life Network, BC Transplant, Quebec Transplant, and other Canadian organ procurement organizations with written informed donor consent as approved by the Human Research Ethics Board at the University of Alberta (Pro00013094). We acknowledge the Life Sciences Institute Imaging Core (RRID:SCR_023783), Life Sciences Institute Flow Cytometry Core and Biomedical Research Centre Sequencing Core facilities at the University of British Columbia for their technical support. The schematics in figures were generated using Biorender.com.

## Author contributions

Conceptualization, methodology and experimental design, J.Z., S.L., and T.J.K.; RNA-seq experiment design and bioinformatic analysis, J.Z., S.L. H.H.C., F.C.L., J.D.J., and T.J.K.; experiment execution and investigation, J.Z., S.L., H.H.C., R.K.B., B.R., and C.Z.; microfluidic chip design, Y.L., H.R., and C.T.; data analysis and statistical quantification, J.Z., S.L., H.H.C., and R.K.B.; original manuscript, J.Z., S.L., H.H.C., and T.J.K.; manuscript review and editing, J.Z., S.L., H.H.C., Y.L., R.K.B., B.R., G.G., C.Z., H.R., C.T., L.C., Y.L., F.C.L., J.D.J., and T.J.K.; supervision of the work, T.J.K.

## Competing interests

T.J.K. was an employee of Fractyl Health during the preparation of this manuscript. The other authors declare no competing interests.
