## [Peer Review File · Nature Communications]

PDX1+ cell budding morphogenesis in a stem cell-derived islet spheroid systemREVIEWER COMMENTS

Reviewer #1 (Remarks to the Author):

Please find my comments the manuscript entitled "PDX1+ cell budding morphogenesis in a stem cell-derived islet spheroid system" by Jia Zhao and colleagues. This report includes data showing a new protocol to generate pancreatic islet cells from hPSCs. For that, the authors refine their previous protocol for endoderm differentiation by fine tuning Wnt signalling. They observed that low Wnt condition allows the production of budding aggregate containing PDX1 cells. These PDX1 progenitors can further differentiate into endocrine cells including Insulin secreting cells. The authors then demonstrate the interest of their system to perform gene function analyses using hPSCs knock out for PDX1 and RFX6. Finally, they uncovered that the Ephrin pathway is involved in the bud formation and the recruitment of PDX1 cells.

This is an interesting study and only few aspects need to be better detailed:

1. How transcriptomically different is the appearance of pancreatic cells in the Wnt med and Wnt low protocols? Wnt low protocol has less PDX1+ cells, but what differs between both protocols in terms of appearance and maturation of pancreatic endocrine and islet cells? Is the morphogenesis/PDX1 rearrangement the 'only' difference?
2. Figure 1SB. The authors report around 5% of PDX1/NKX6.1 cells in stage 4 (text and b flow cytometry), however, on the immunostaining showed on this figure for stage 4, NKX6.1 positive cell are difficult to find. Can the authors provide a better staining?
3. How efficient is bud formation? Do all aggregate under Wnt low conditions form a PDX1+ bud?
4. Similarly, the authors claim that their protocol can work on multiple hPSCs. However, there is little information about efficacy of bud formation. Can they show staining showing bud formation with each cell line and provide the efficiency of bud formation.
5. Figure S2B. Looking at flow cytometry data for PDX1 and NKX6.1 it looks like the cell lines tested do not activate NKX6.1 at stage 4? This is in contrast of graph S2D where it looks like there are NKX6.1 positive cells.
6. Can the authors show the staining for the whole cluster of Figure 1E, not only the islet bud?
7. The high enrichment of INS cells in localized bud structures seem to also happen in stage 7 wnt med condition (figure S1A)? They authors should discuss in more details why this new system could be advantageous compared to previous method.
8. Several times the authors refer to data not shown. Can they explain why or show the data in supplementary figures?
9. Figure 5E: the genes are very difficult to correlate with the lines in the heatmap. The authors can maybe divide the heatmap in different parts so that the names of the genes can be better annotated.
10. Figure 5H: some qPCR gene expression data does not correlate with the RNAseq data showed in heatmap for the same genes. For example KRT18 is showed as not significantly expressed by qPCR, but seem to have a very big difference in gene expression on heatmap between bodies and buds... Could the authors find a better way of plotting qPCR data such as relative gene expression without fold change comparison so that the real gene expression levels can be better visualized? Human islet

controls can be plotted next to the experimental conditions if needed.

11. Figure 6J does not show a striking change in hormonal cells thereby suggesting that migration of PDX1 cells and budding might not be essential. It would be very useful to demonstrate that Ephrin inhibition increases/decreases the number of multi hormonal cells. This could show that their budding system and the PDX1 migration is indeed necessary to produce functional endocrine cells.

Reviewer #2 (Remarks to the Author):

In this manuscript, Kieffer and colleagues developed a strategy for generating stem cell-derived islet spheroids that incorporates a PDX1+ cell budding step in the differentiation protocol. The 'budding-type differentiation' was achieved by lowering the level of Wnt signaling in culture at the endoderm differentiation stage. The authors showed that this differentiation strategy leads to a core-mantle arrangement of the islet cells, with the β -cells in a core position while α -cells in the mantle, and an enhancement of islet cell maturation. In addition, the EphB3/4 signaling was found to be involved in PDX1+ cell budding morphogenesis. Based on the observations presented in this study, the authors proposed that the PDX1+ budding step mimics the in vivo islet morphogenesis process, whereby PDX1+ progenitor cells would first cluster and subsequently develop into islet cells. This model resembles the way principal islets are formed in lower vertebrates, such as fish and lamprey models.

The question investigated in this paper is interesting and relevant for the islet biology as well as diabetes cell therapy fields. Indeed, learning how to reproduce islet morphogenesis in the differentiation protocols is needed for improving the functional maturity of stem cell-derived islets. However, there are a number of issues that need to be addressed here to support the conclusions drawn by the authors. I listed below specific concerns with the manuscript:

- 1) The authors made use of the sentence 'data not shown' multiple times in the manuscript (see pp. 3, 6, 7, Results section). This is not acceptable, the data should be always included in the Main or Supplementary figures. Especially, some of these 'data not shown' results are particularly relevant being either control experiments (e.g. Wnt target gene modulation should be shown as control of an effective Wnt low/med strategy) or evidence directly in support of major conclusions (e.g. evidence for Robo/ Rho-actin/ Wnt signalings not affecting PDX1+ cell clustering should be shown and not reported as 'data not shown').
- 2) In Figure 1E, hormone+ cells are not clustered as shown in Fig. 1 C and D with the INS-GFP line. An important control missing here is a co-staining of the INS-GFP-derived clusters with an anti-insulin antibody and anti-GFP to show the overlay between the two signals.
- 3) The movies performed with the PDX1-reporter hESC line in support of the migration of PDX1+ cells need to be included in the paper.
- 4) The conclusion drawn from the analysis of the PDX1-KO hESCs phenotype should be revised. It is known that in the absence of PDX1 pancreatic fate specification does not occur; budding morphogenesis cannot start here as there are no pancreatic progenitors in the culture. Thus, the statement '... expression of PDX1 induces cells to undergo budding morphogenesis in our spheroid system' (see p. 5, Results section) is wrong and needs to be revised. The experiments performed on PDX1-KO line could be entirely removed, as it is inconclusive. Actually, the results obtained with the RFX6-KO line suggest that high /sustained levels of PDX1 are required to initiate the clustering of the pancreatic progenitors and, possibly, subsequent islet morphogenesis. This would be in line with the well-known increase of PDX1 levels in vivo in the trunk / future endocrine cells at later stages after pancreatic specification. Thus, an hypomorphic allele or temporally controlled mutation would be more appropriate here instead of using a PDX1 KO line.

5) The characterization of the two compartments 'islets buds and main bodies' needs to be expanded. Bulk RNASeq and RT-qPCR validation showed a set of differentially expressed genes between the islet buds and main bodies, however, based on both analyses one cannot exclude the presence cell type mixtures in each compartment. For instance, RT-qPCR (see Figs. 5I-K) showed fibroblast markers (THY1, COLA1, VIM) being expressed also in GFP+ islets buds and not exclusively in the main bodies. Thus, one cannot rule out the presence of fibroblasts in the islets bud compartment and their potential role in islet morphogenesis/differentiation and maturation. The same applies to exocrine marker genes. The authors should characterize the main differences between the 2 compartments at least by immunostaining, as they did for the Ephrin signaling molecules (see Fig. S7).

Moreover, in this paper the 2 compartments were separated at stage S6, which is quite late in the differentiation protocol; thus, one can only conclude that surrounding cues/cell-cell interactions are not required for stem cell-derived islet maturation but cannot exclude an earlier role in differentiation and morphogenesis. This should be rectified in the manuscript. Alternatively, the experiment should be designed in a different way and includes the separation of the 2 compartments at earlier time points.

6) The analysis of NGN3 expression should be examined upon EphB signaling inhibition and added to complete the analysis in Fig. 6 I,J.

7) The model proposed here is unclear. It is well established that all pancreatic progenitors once specified are PDX1+ cells. Thus, according to their model, PDX1+ cells should all undergo clustering and, subsequently, give rise to islet cells. This would mean that an entire pancreatic bud would become an islet, which is unlikely. Perhaps, there is a certain heterogeneity among PDX1+ pancreatic progenitors and the difference between a PDX1+ pancreatic progenitor that becomes an islet cell or not lies in the PDX1 levels. This seems a more likely model.

Also, the legend of Figure 7 accompanying the model should be revised. The statement '...we observe PDX1+ cell budding morphogenesis prior to pancreatic specification ...' is inaccurate. PDX1 is a transcription factor required for pancreatic fate specification. Perhaps, the authors meant 'prior to endocrine specification'?

8) A core-mantle islet architecture is typical of the mouse islets, while in human the islets cells are scattered and intermingled. Based on this study, it seems that the murine core-mantle islet architecture performs better than the human one in a stem cell culture model. The authors should discuss these aspects in the Discussion section.

Reviewer #3 (Remarks to the Author):

In this manuscript, Zhao and Liang et al demonstrate a new protocol for hESC modelling of pancreas budding morphogenesis. They demonstrate that Wnt concentrations at the beginning of the differentiation protocol is critical for inducing the PDX1+ dependent budding morphogenesis model and they validate their findings using RNA-Seq of the buds vs main body of the spheroids. They use the model to study the role of PDX1-RFX6 transcription factors in budding human morphogenesis and also demonstrate a role for Ephrin signaling in the budding morphogenesis process. The manuscript is clear, well written and the importance of a human model for embryonic human pancreas budding morphogenesis is of interest to the scientific community. While most of the work is convincing there are certain points that could be further clarified:

Major points:

- The classification and characterization of the Wntmed and Wntlow should be more clear in the manuscript. The range of the Chir concentrations to separate these two conditions are quite narrow. I would suggest that a range of concentration necessary for bud formation needs to be tested and clearly reported in the figure to make the protocol clear, i.e what is a low concentration of Chir that bud is no longer formed?
- How efficient and reproducible is the bud formation during the differentiation process with the new

protocol? Please report the percentage of clusters that form a bud during the differentiation. Is there any spontaneous bud formation observed with the Wntmed conditions?

- Is there any extracellular matrix coating used during the differentiation? Please include this information as it will be necessary for the reproduction of the protocol
- Does Pdx1 directly regulates Ephrin signaling and expression? It would be interesting to check the Pdx1 KO line for expression of Ephrin proteins.
- To support further the model and connect with the literature of budding morphogenesis that the authors discuss, it would be interesting to check how the bud "connects" to the main body. Please stain for adhesion molecules e.g Ecadherin-integrin molecules-EMT markers and cell polarity markers to get a more clear insight into the budding process.

Minor points:

- Assess directly the PDX1 proliferation rate (e.g EdU incorporation and FACS) during bud formation.
- Some of the data not shown parts would be informative to be included for example:

1. Wnt target genes

2. The screen of the individual components of stage 7 media

3. The data on the screen of different signaling inhibitors on page 6

- If feasible in Supplementary figure 7 validate ephrin expression by western blot
- For completeness in the zebrafish references please cite this work for ephrin signaling in liver bud formation: PMID: 27825440 and this reference that first showed bud formation in zebrafish larva: PMID: 12941629

- A recent preprint used Rfx6 KO iPSC system in the pancreas context, please discuss in relation to your findings <https://doi.org/10.1101/2023.11.15.567202>

Reviewer #1 (Remarks to the Author):

Please find my comments the manuscript entitled “PDX1+ cell budding morphogenesis in a stem cell-derived islet spheroid system” by Jia Zhao and colleagues. This report includes data showing a new protocol to generate pancreatic islet cells from hPSCs. For that, the authors refine their previous protocol for endoderm differentiation by fine tuning Wnt signalling. They observed that low Wnt condition allows the production of budding aggregate containing PDX1 cells. These PDX1 progenitors can further differentiate into endocrine cells including Insulin secreting cells. The authors then demonstrate the interest of their system to perform gene function analyses using hPSCs knock out for PDX1 and RFX6. Finally, they uncovered that the Ephrin pathway is involved in the bud formation and the recruitment of PDX1 cells. This is an interesting study and only few aspects need to be better detailed:

Response: We thank the reviewer for their positive assessment of our work and for the suggestions below on how to improve upon our study. We have performed additional experiments and analyses following your suggestions. The manuscript text has been extensively revised (highlighted with yellow background), and now contains new data within main figures plus 11 new supplementary figures and supplementary tables. Below please find our detailed point-to-point responses in blue font.

1. How transcriptomically different is the appearance of pancreatic cells in the Wnt med and Wnt low protocols? Wnt low protocol has less PDX1+ cells, but what differs between both protocols in terms of appearance and maturation of pancreatic endocrine and islet cells? Is the morphogenesis/PDX1 rearrangement the ‘only’ difference?

Response: We thank the reviewer for raising these questions. To better assess the differences in the cells generated by the Wnt-med and Wnt-low protocols, we have performed the following experiments: (1) RT-qPCR assays to examine a panel of key transcripts at multiple differentiation stages (**Supplementary Fig. 3**); and (2) static glucose stimulated insulin secretion (GSIS) assays to assess the functionality of sorted INS+ beta cells at Stage 7 (**Supplementary Fig. 1h**). We find that pancreatic cell transcripts (PDX1, NKX6.1, SOX9) and pro-endocrine cell transcripts (NGN3, NEUROD1) were lower in budding type Stage 4 cells generated in Wnt low conditions than cells obtained from the Wnt medium bulk-type differentiation condition (**Supplementary Fig. 3b**). However, the expression levels of most transcripts became comparable between budding type and bulk type cells at Stages 5 and 7 (**Supplementary Fig. 3c-3d**). Nevertheless, relative to bulk type cells, we notice higher levels of GHRL, SLC18A1, KCNK1 transcripts and lower levels of GCG, PPY, ARX, NKX2.2, ABCC8 transcripts in the budding type Stage 7 cells (**Supplementary Fig. 3d**). Although transcripts of functional beta cell markers MAFA, IAPP, GCK, PSCK1 and KCNK3 were similar between the two types of differentiation (**Supplementary Fig. 3d**), insulin secretion and total insulin content of budding type Stage 7 beta cells was 1.4-fold and 1.7-fold lower than those of bulk type cells, respectively (**Supplementary Fig. 1h**), suggesting a less mature phenotype of beta cells generated by the budding protocol. To summarize these findings, we also created a new supplementary table that highlights key differences of the cells generated by the two protocols (**Supplementary Table 1**). We have included these new results in the revised manuscript (**Page 3**).

2. Figure S1B. The authors report around 5% of PDX1/NKX6.1 cells in stage 4 (text and b flow cytometry), however, on the immunostaining showed on this figure for stage 4, NKX6.1 positive cell are difficult to find. Can the authors provide a better staining?

Response: Thank you for this suggestion. We have now replaced this image with a more representative immunostaining image that shows the detection of a few NKX6.1-positive cells in the budding type stage 4 cells (**Supplementary Fig. 1e**).

3. How efficient is bud formation? Do all aggregates under Wnt low conditions form a PDX1+ bud?

Response: Thank you for raising these important questions. We have now added the quantifications of bud formation efficiencies (including PDX1+ buds and INS+ islet buds) during the differentiation process with our Wnt low protocol on five hPSC lines tested in this study. The percentages of clusters that form PDX1+ bud(s) for each cell line are 98.7% ± 0.2% (HUES4 PDXeG), 93.1% ± 3.1% (Mel1 INS^{GFP/W}), 88.3% ± 0.9% (H1 hESC), 87.7% ± 1.4% (GCaMP hiPSC), and 97.3% ± 0.4% (HUES8) (**Supplementary Fig. 5a-5b, 5d**). The percentages of clusters that form INS+ islet bud(s) quantified at the end of protocol for each cell line are 92.5% ± 4.1% (HUES4 PDXeG), 85.4% ± 3.2% (Mel1 INS^{GFP/W}), 77.4% ± 10.2% (H1 hESC), 71.4% ± 12.3% (GCaMP hiPSC), and 84.5% ± 5.9% (HUES8) (**Supplementary Fig. 5e-5f, 5h**). Moreover, the number of PDX1+ or INS+ buds per main body was quantified in HUES4 PDXeG and Mel1 INS^{GFP/W} reporter lines, respectively, showing > 75% main bodies with only one bud and ~ 10% main bodies with two or more buds (**Supplementary Fig. 5c, 5g**). Collectively, these results demonstrate the high efficiency and reproducibility of bud formation observed with the Wnt low protocol in various stem cell lines. We have added these new results in the revised manuscript (**Supplementary Fig. 5 and Pages 3-4**).

4. Similarly, the authors claim that their protocol can work on multiple hPSCs. However, there is little information about efficacy of bud formation. Can they show staining showing bud formation with each cell line and provide the efficiency of bud formation.

Response: Please see our detailed responses above to Comment #3. Our new data reveal high efficiency and reproducibility of bud formation observed with the Wnt low protocol in various stem cell lines (**Supplementary Fig. 5 and Pages 3-4**). Representative immunostaining and live cell images showing bud formation from each cell line are included in the revised manuscript (**Fig. 1c and Supplementary Figs. 4e-4f, 5 and 11**).

5. Figure S2B. Looking at flow cytometry data for PDX1 and NKX6.1 it looks like the cell lines do not activate NKX6.1 at stage 4? This is in contrast of graph S2D where it looks like there are NKX6.1 positive cells.

Response: In Figure S2B (now **Supplementary Fig. 4b**, representative flow cytometry data) and Figure S2D (now **Supplementary Fig. 4d**, the quantification), we show that NKX6.1+ cells can be induced at stage 4 but at extremely low percentages. On average, 0.5%-3% NKX6.1+ cells are detected at stage 4 in the cell lines we tested.

6. Can the authors show the staining for the whole cluster of Figure 1E, not only the islet bud?

Response: Indeed, Figure 1E shows immunostaining of an isolated stage 7 islet bud, to highlight that the islet bud compartment contains major islet cell types. Nevertheless, as requested we have now added a supplementary figure (**Supplementary Fig. 6**) with several examples showing the staining for whole clusters with both compartments of islet bud and main body.

7. The high enrichment of INS cells in localized bud structures seem to also happen in stage 7 wnt med condition (figure S1A)? They authors should discuss in more details why this new system could be advantageous compared to previous method.

Response: Local enrichment of INS+ cells may be seen in stage 7 Wnt-med conditions, which is in line with similar observations in other publications (PMID: 35241836 and PMID: 34380694). This may represent the process of islet cell rearrangement at maturing stages in developing human islets. By contrast, our budding model emphasizes the budding morphogenesis of PDX1+ progenitor cells, recapitulating pancreatic bud formation, and an event occurring at an earlier developmental stage.

Compared to previous methods, this new system provides a unique human islet developmental model to study asynchronous differentiation within a heterogeneous cell-cell interaction environment, whereby pancreatic cell sorting, tissue segregation and islet morphogenesis occur. This approach may complement

the use of human fetal pancreas tissues, mouse organoid models and other tools using rodents. We now mention these points in the Discussion (Page 8) and have included a new summary table (Supplementary Table 1) highlighting key differences and utility/advantages of this new system compared to previous methods.

8. Several times the authors refer to data not shown. Can they explain why or show the data in supplementary figures?

Response: Thank you for raising this concern. We have now added all these data in new supplementary figures to improve the transparency of our results. These include: (1) qPCR 96-well plate array data showing differential Wnt pathway genes by Wnt-med and Wnt-low conditions (Supplementary Fig. 2); (2) immunostaining data showing the screen of individual components of stage 7 media on islet architecture (Supplementary Fig. 9); and (3) live cell fluorescent imaging data showing the screen of different signaling pathways on PDX1+ cell clustering (Supplementary Fig. 14).

9. Figure 5E: the genes are very difficult to correlate with the lines in the heatmap. The authors can maybe divide the heatmap in different parts so that the names of the genes can be better annotated.

Response: Thank you for this helpful suggestion to improve clarity. We have created a new heatmap highlighting the selective genes of interests in Fig. 5f to improve data visualization such that the names of genes and the varied expression of a particular gene across different samples are now better annotated.

10. Figure 5H: some qPCR gene expression data does not correlate with the RNA-seq data showed in heatmap for the same genes. For example, KRT18 is showed as not significantly expressed by qPCR, but seem to have a very big difference in gene expression on heatmap between bodies and buds... Could the authors find a better way of plotting qPCR data such as relative gene expression without fold change comparison so that the real gene expression levels can be better visualized? Human islet controls can be plotted next to the experimental conditions if needed.

Response: We thank this reviewer for pointing out the differences in qPCR and RNA-seq plots. The differences in RNA-seq heatmap and qPCR plots are partly due to the different techniques and normalization methods. In RNA-seq, gene counts after DESeq2 normalization and variance stabilizing transformation ($\sim \log_2$ scale) are converted to row z-scores to better illustrate the differences across samples on heatmap. qPCR data were first normalized to a housekeeping gene (NFX1), and then the relative gene expression was compared to the expression level in human islets. In the case of KRT18, the RNA-seq data ($\sim \log_2$ scale) of body and bud are 8.84 ± 0.06 versus 8.27 ± 0.10 (t-test $p = 0.01$), and their z-scores (after row z-score scaling) makes the difference look more striking on heatmaps. The expressions of KRT18 in qPCR are 0.32 ± 0.09 in body and 0.15 ± 0.03 in bud (t-test $p = 0.07$). Consistently, both techniques show that body cells tend to express higher level of KRT18 transcripts relative to bud cells.

We agree with this reviewer that relative gene expression levels are more informative than fold change comparison. Following this suggestion, we have plotted our qPCR data as gene expression levels relative to human islets in the revised Fig. 5i-5l.

11. Figure 6J does not show a striking change in hormonal cells thereby suggesting that migration of PDX1 cells and budding might not be essential. It would be very useful to demonstrate that Ephrin inhibition increases/decreases the number of multi hormonal cells. This could show that their budding system and the PDX1 migration is indeed necessary to produce functional endocrine cells.

Response: The Figure 6J (now is Fig. 6k) showed statistically significant changes in the percentages of hormonal cells after EphB inhibition (INS+ cells: $43\% \pm 2.6\%$ in DMSO versus $19.5\% \pm 3.7\%$ in EphB3/4 co-inhibition; and GCG+ cells: $18.7\% \pm 1.2\%$ in DMSO versus $8.1\% \pm 1.6\%$ in EphB3/4 co-inhibition). As per

suggestion from Reviewer #2, we also added new data in Figure 6I showing that NGN3 expression was significantly reduced after EphB inhibition ($6.7\% \pm 0.8\%$ in DMSO versus $1.7\% \pm 0.4\%$ in EphB3/4 co-inhibition). Collectively, these results support our conclusion that PDX1 cell clustering is an important step to produce subsequent endocrine precursors and islet cells (Fig. 6). We have now clarified these results in the revised manuscript (Page 8). Nevertheless, we are aware that the EphB3/4 co-inhibition treatment may not completely block EphB signaling (Supplementary Figs. 14 and 16), and we have discussed this limitation in our revised manuscript (Page 9).

Reviewer #2 (Remarks to the Author):

In this manuscript, Kieffer and colleagues developed a strategy for generating stem cell-derived islet spheroids that incorporates a PDX1+ cell budding step in the differentiation protocol. The ‘budding-type differentiation’ was achieved by lowering the level of Wnt signaling in culture at the endoderm differentiation stage. The authors showed that this differentiation strategy leads to a core-mantle arrangement of the islet cells, with the β -cells in a core position while α -cells in the mantle, and an enhancement of islet cell maturation. In addition, the EphB3/4 signaling was found to be involved in PDX1+ cell budding morphogenesis. Based on the observations presented in this study, the authors proposed that the PDX1+ budding step mimics the in vivo islet morphogenesis process, whereby PDX1+ progenitor cells would first cluster and subsequently develop into islet cells. This model resembles the way principal islets are formed in lower vertebrates, such as fish and lamprey models.

Response: We thank this reviewer for their time assessing our manuscript and for the thoughtful summary of our findings.

The question investigated in this paper is interesting and relevant for the islet biology as well as diabetes cell therapy fields. Indeed, learning how to reproduce islet morphogenesis in the differentiation protocols is needed for improving the functional maturity of stem cell-derived islets. However, there are a number of issues that need to be addressed here to support the conclusions drawn by the authors. I listed below specific concerns with the manuscript:

Response: We appreciate the supportive comments from this reviewer. We have now addressed these concerns by performing additional experiments and adding new data, analysis, and relevant discussions. The manuscript text has been extensively revised (highlighted with yellow background), and now contains new data within main figures plus 11 new supplementary figures and supplementary tables. Point-to-point responses are detailed below to address each specific concern.

1) The authors made use of the sentence ‘data not shown’ multiple times in the manuscript (see pp. 3, 6, 7, Results section). This not acceptable, the data should be always included in the Main or Supplementary figures. Especially, some of these ‘data not shown’ results are particularly relevant being either control experiments (e.g., Wnt target gene modulation should be shown as control of an effective Wnt low/med strategy) or evidence directly in support of major conclusions (e.g., evidence for Robo/ Rho-actin/ Wnt signaling not affecting PDX1+ cell clustering should be shown and not reported as ‘data not shown’).

Response: We thank the reviewer for raising this concern. We have now added all these data in new supplementary figures to improve the transparency of our results. These include: (1) qPCR 96-well plate array data showing differential Wnt pathway genes by Wnt-med and Wnt-low conditions (Supplementary Fig. 2); (2) immunostaining data showing the screen of individual components of stage 7 media on islet architecture (Supplementary Fig. 9); and (3) live cell fluorescent imaging data showing the screen of different signaling pathways on PDX1+ cell clustering (Supplementary Fig. 14).

2) In Figure 1E, hormone+ cells are not clustered as shown in Fig. 1 C and D with the INS-GFP line. An

important control missing here is a co-staining of the INS-GFP-derived clusters with an anti-insulin antibody and anti-GFP to show the overlay between the two signals.

Response: We are sorry for the lack of clarity here. Figure 1E shows immunostaining of an isolated stage 7 islet bud and it intends to highlight that the islet bud compartment contains the major islet cell types. We have now added a supplementary figure (**Supplementary Fig. 6**) with several examples showing the staining for whole clusters with both compartments of islet bud and main body.

We agree with the reviewer that it is important to demonstrate the reliability of the INS-GFP reporter line that we used in this study. We have performed both immunostaining and flow cytometry analysis, showing 87.1% (from n = 2 independent experiments) overlay of GFP signals by anti-GFP antibody and insulin signals by anti-insulin antibody (**Supplementary Fig. 19a-19c**). Similarly, the PDX1 reporter line (HUES4 PDXeG) was validated by imaging GFP reporter signals and PDX1 signals by anti-PDX1 antibody, illustrating 89.1% (from n = 2 independent experiments) overlay of signals (**Supplementary Fig. 19d-19f**). We have added these data and specified this information in the revised manuscript (**Page 12**).

3) The movies performed with the PDX1-reporter hESC line in support of the migration of PDX1+ cells need to be included in the paper.

Response: We apologize for giving the readers the impression that time-lapse movies were generated. Rather, these data obtained with the PDX1 reporter hESC line are time-course snapshot images (**Figs. 2d, 6d and Supplementary Figs. 5, 8**) rather than timelapse images. To track the process of PDX1+ cell migration in living cultures, we differentiated individual clusters in ultralow attachment U-bottom 96-wells and captured several images from each well with a conventional fluorescent microscope on a daily basis. We have clarified this information in the revised manuscript (**Page 4**). Although we do not have a suitable imager capable of doing long-term timelapse imaging, we feel that our findings still support our conclusion of PDX1+ cell migration and clustering.

4) The conclusion drawn from the analysis of the PDX1-KO hESCs phenotype should be revised. It is known that in the absence of PDX1 pancreatic fate specification does not occur; budding morphogenesis cannot start here as there are no pancreatic progenitors in the culture. Thus, the statement ‘... expression of PDX1 induces cells to undergo budding morphogenesis in our spheroid system’ (see p. 5, Results section) is wrong and needs to be revised. The experiments performed on PDX1-KO line could be entirely removed, as it is inconclusive. Actually, the results obtained with the RFX6-KO line suggest that high /sustained levels of PDX1 are required to initiate the clustering of the pancreatic progenitors and, possibly, subsequent islet morphogenesis. This would be in line with the well-known increase of PDX1 levels in vivo in the trunk / future endocrine cells at later stages after pancreatic specification. Thus, an hypomorphic allele or temporally controlled mutation would be more appropriate here instead of using a PDX1 KO line.

Response: Thank you for these constructive comments. We fully acknowledge that since knockout of PDX1 disrupts the formation of the pancreatic progenitor pool, it is not possible to determine the role of PDX1 in downstream events from analysis of PDX1 knockout cells. We now ensure that in the revised manuscript, we separate our conclusion regarding the role of PDX1 in budding morphogenesis from studies with the PDX1 knockout cells. Specifically, we have removed the concluding sentence “expression of PDX1 induces cells to undergo budding morphogenesis in our spheroid system” in this result section, and changed the concluding remarks to “These results corroborate previous work by recapitulating the pancreatic agenesis phenotype with a stem cell model and again emphasize the importance of PDX1 in pancreatic fate specification and morphogenesis” in the revised manuscript.

We rely on several other observations to support the role of PDX1 in budding morphogenesis, including as the reviewer notes, our interesting findings with the RFX6-KO line demonstrating that the sustained

expression of PDX1 is required to initiate the clustering of pancreatic progenitors and subsequent islet cell morphogenesis in a heterogeneous cellular environment (i.e., containing both PDX1-expressing cells and PDX1-negative cells). Our conclusion is also supported by our live cell tracking experiments in PDX1 reporter hESC cultures (**Fig. 2d and Supplementary Fig. 8a-8i**). Specifically, with the use of a hypomorphic PDX1-EGFP reporter line (i.e., one PDX1 allele is replaced by EGFP sequence), we provided direct evidence that pancreatic budding morphogenesis is mediated by spontaneous clustering of PDX1+ cells. While experiments performed with the PDX1 KO line cannot inform the role of PDX1 beyond pancreatic progenitor formation, the studies still illustrate the utility of our stem cell systems in modelling human pancreas diseases.

5) The characterization of the two compartments 'islets buds and main bodies' needs to be expanded. Bulk RNA-Seq and RT-qPCR validation showed a set of differentially expressed genes between the islet buds and main bodies, however, based on both analyses one cannot exclude the presence cell type mixtures in each compartment. For instance, RT-qPCR (see Figs. 5I-K) showed fibroblast markers (THY1, COLA1, VIM) being expressed also in GFP+ islets buds and not exclusively in the main bodies. Thus, one cannot rule out the presence of fibroblasts in the islets bud compartment and their potential role in islet morphogenesis/differentiation and maturation. The same applies to exocrine marker genes. The authors should characterize the main differences between the 2 compartments at least by immunostaining, as they did for the Ephrin signaling molecules (see Fig. S7). Moreover, in this paper the 2 compartments were separated at stage S6, which is quite late in the differentiation protocol; thus, one can only conclude that surrounding cues/cell-cell interactions are not required for stem cell-derived islet maturation but cannot exclude an earlier role in differentiation and morphogenesis. This should be rectified in the manuscript. Alternatively, the experiment should be designed in a different way and includes the separation of the 2 compartments at earlier time points.

Response: Thank you for these constructive comments. Guided by our own bulk RNA-seq and qPCR data, we have now immunostained several differentially expressed candidate genes, including THY1, VIM, SOX9, YAP1 and SOX2 (an anterior foregut marker), to further characterize main body cells as per the suggestion (**Supplementary Fig. 13**). Specifically, we show (1) remarkably higher expression levels of THY1 and VIM (pancreatic mesenchyme or fibroblast markers) in the main bodies; (2) mutually exclusive expression of the ductal cell marker SOX9 in main body cells and the endocrine cell marker NEUROD1 in bud cells; (3) higher expression level of YAP1 (a mechanosensitive signal for balancing progenitor cell self-renewal and differentiation) in the main body cells versus lower expression of YAP1 in the NEUROD1+ endocrine-committed bud cells; and (4) higher expression of the anterior foregut marker SOX2 in main body cells. We also examined the exocrine cell marker Trypsin 1/2/3 but did not detect any expression in either bud or body cells (**Supplementary Fig. 13**), ruling out the exocrine identity of main body cells. Collectively, these results suggest a potential mesenchyme/fibroblast phenotype or other lineage commitment (ductal or anterior foregut) but not an exocrine lineage of the main body cells. We have now added these results in the revised manuscript (**Supplementary Fig. 13 and Page 7**).

We agree with this reviewer that we cannot exclude an earlier role of surrounding cues from main body cells in islet bud morphogenesis/differentiation. Due to technical limitation, we are only able to separate the two compartments at early stage 6 (e.g., S6D3) when the buds are more protruding from main bodies. We have discussed this limitation in our revised manuscript (**Pages 7 and 11**).

6) The analysis of NGN3 expression should be examined upon EphB signaling inhibition and added to complete the analysis in Fig. 6 I, J.

Response: Thank you for this suggestion. Although we are still working on protocol optimization for assaying NGN3 expression under flow cytometry, we have successfully examined NGN3 expression by

immunostaining to complete the analysis in Figure 6I-6J (now is Figure 6J-6K). By normalization to the total PDX1+ cells, we show that NGN3 expression is significantly impaired upon EphB3/4 co-inhibition compared to DMSO control ($1.7\% \pm 0.4\%$ in EphB3/4 co-inhibition versus $6.7\% \pm 0.8\%$ in DMSO). We have added these data in the new **Fig. 6i** and discussed these results in the revised manuscript (**Page 8**).

7) The model proposed here is unclear. It is well established that all pancreatic progenitors once specified are PDX1+ cells. Thus, according to their model, PDX1+ cells should all undergo clustering and, subsequently, give rise to islet cells. This would mean that an entire pancreatic bud would become an islet, which is unlikely. Perhaps, there is a certain heterogeneity among PDX1+ pancreatic progenitors and the difference between a PDX1+ pancreatic progenitor that becomes an islet cell or not lies in the PDX1 levels. This seems a more likely model. Also, the legend of Figure 7 accompanying the model should be revised. The statement ‘...we observe PDX1+ cell budding morphogenesis prior to pancreatic specification ...’ is inaccurate. PDX1 is a transcription factor required for pancreatic fate specification. Perhaps, the authors meant ‘prior to endocrine specification’?

Response: Thank you for these constructive comments. We demonstrate that a vast majority of PDX1+ cells undergo clustering and budding out from PDX1-negative main bodies prior to endocrine specification (**Fig. 2b, 2d and Supplementary Figs. 5, 8, 15**) in our budding type differentiation system. With our media highly enriched with endocrine cell-directed cocktails (which inhibits progenitor cell proliferation and does not support ductal or exocrine cell commitment) at later stages, the PDX1+ pancreatic bud efficiently gives rise to an endocrine islet (**Fig. 2a**). This paradigm is supported by observations of islet formation in fish and lamprey, where a dorsal bud is formed through the convergence of PDX1+ cells that later develop into a principal islet (PMIDs: 30133710, 22034951, 11377827 and 26435359). Nevertheless, we note that only a certain population of PDX1+ progenitor cells, very likely the ones with high/peak PDX1 expression level (**Fig. 2g**), are specified to islet cells within the bud structure (**Figs. 2a, 2f-2h, 4e**), in line with this reviewer’s comments. We do not wish to give readers the impression that all PDX1+ cells become islet cells. The model proposed in the original schematic of Fig. 7a intended to show that the bud structure formed by PDX1+ cell clustering creates a favorable niche and subsequently promotes local pro-endocrine cell and islet cell commitment. We have now revised Fig. 7a to show the partial endocrine commitment and made further clarification in the legend of Fig. 7 (**Page 20**). We also have corrected “prior to pancreatic specification” to “prior to endocrine specification” in our revised manuscript.

8) A core-mantle islet architecture is typical of the mouse islets, while in human the islets cells are scattered and intermingled. Based on this study, it seems that the murine core-mantle islet architecture performs better than the human one in a stem cell culture model. The authors should discuss these aspects in the Discussion section.

Response: We thank the reviewer for raising this important observation. In our budding type differentiation model, the fact that stage 7 islet buds display better functionality than stage 6 islet buds may emphasize that both architecture and cellular maturity states (e.g., mono- or bi-hormonal) determine islet function. Indeed, we show the majority of stage 6 islet cells are INS+/GCG+ bi-hormonal cells adopting an intermingled structure, whereas stage 7 islet cells are mostly INS+/GCG- and INS-/GCG+ monohormonal cells with a favorable core-mantle organization (**Fig. 3a-3c**). Moreover, the core-mantle architecture renders beta cells with more homotypic cell-cell interactions, which is associated with more synchronized calcium activities under glucose stimulation (**Fig. 3e-3g**), providing a plausible explanation that the core-mantle islet architecture performs better than a mixed one in our stem cell model. In addition, the Susan Bonner-Weir group reported that small-sized human islets prefer an adoption of core-mantle pattern (like mouse islets) and large-sized islets with a mixed organization seem derived from small “core-mantle modular units” coalescing (PMID: 25604813), suggesting that cellular rearrangement occurs in human islets. We have incorporated this discussion into our revised manuscript (**Page 10**).

Reviewer #3 (Remarks to the Author):

In this manuscript, Zhao and Liang et al demonstrate a new protocol for hESC modelling of pancreas budding morphogenesis. They demonstrate that Wnt concentrations at the beginning of the differentiation protocol is critical for inducing the PDX1+ dependent budding morphogenesis model and they validate their findings using RNA-Seq of the bud vs main body of the spheroids. They use the model to study the role of PDX1-RFX6 transcription factors in budding human morphogenesis and also demonstrate a role for Ephrin signaling in the budding morphogenesis process. The manuscript is clear, well written and the importance of a human model for embryonic human pancreas budding morphogenesis is of interest to the scientific community. While most of the work is convincing there are certain points that could be further clarified:

Response: We thank this reviewer for the thorough consideration of our work and appreciate their positive assessment of our studies. We have performed additional experiments and analyses following your suggestions. The manuscript text has been extensively revised (highlighted with yellow background), and now contains new data within main figures plus 11 new supplementary figures and supplementary tables. Below please find our detailed point-to-point responses in blue font.

Major points:

- The classification and characterization of the Wnt med and Wnt low should be much clear in the manuscript. The range of the Chir concentrations to separate these two conditions are quite narrow. I would suggest that a range of concentration necessary for bud formation needs to be tested and clearly reported in the figure to make the protocol clear, i.e., what is a low concentration of Chir that bud is no longer formed?

Response: Thank you for this suggestion. Indeed, the concentrations of Wnt agonists (CHIR, MCX, Wnt3a) used to separate the two types of differentiation are fine-tuned and cell line dependent. Specifically, based upon our extensive dose-ranging studies (new data added in **Supplementary Fig. 1a-1c** and a summary of Wnt dose conditions shown in **Supplementary Fig. 4a**), bulk type differentiation is switched on when CHIR is used at 3 μ M in combination with GDF8 or Activin A. Thus, we use a minimum of 3 μ M CHIR (or 1 μ M MCX) to induce bulk type differentiation (i.e., PDX1+ budding is no longer formed under this condition). Depending on the cell lines used, budding type differentiation is induced when CHIR is at 0.2-1.5 μ M (or MCX is used at 0.2-0.5 μ M). We have now specified the CHIR or MCX concentrations used to induce bulk or budding type differentiation in each figure legend. We have also summarized the range of CHIR, MCX and Wnt3a concentrations for inducing budding in the cell lines tested in this study in the revised **Supplementary Fig. 4a**.

- How efficient and reproducible is the bud formation during the differentiation process with the new protocol? Please report the percentage of clusters that form a bud during the differentiation. Is there any spontaneous bud formation observed with the Wntmed conditions?

Response: Thank you for these important questions. We have now added the quantifications of bud formation efficiencies (including PDX1+ buds and INS+ islet buds) during the differentiation process with our Wnt low protocol on five hPSC lines tested in this study. Specifically, the percentages of clusters that form PDX1+ buds for each cell line are 98.7% \pm 0.2% (HUES4 PDXeG), 93.1% \pm 3.1% (Mel1 INS^{GFP/W}), 88.3% \pm 0.9% (H1 hESC), 87.7% \pm 1.4% (GCaMP hiPSC), and 97.3% \pm 0.4% (HUES8) (**Supplementary Fig. 5a-5b, 5d**). The percentages of clusters that form INS+ islet buds quantified at the end of protocol for each cell line are 92.5% \pm 4.1% (HUES4 PDXeG), 85.4% \pm 3.2% (Mel1 INS^{GFP/W}), 77.4% \pm 10.2% (H1 hESC), 71.4% \pm 12.3% (GCaMP hiPSC), and 84.5% \pm 5.9% (HUES8) (**Supplementary Fig. 5e-5f, 5h**). Moreover, the number of PDX1+ or INS+ buds per main body was quantified in HUES4 PDXeG and Mel1 INS^{GFP/W} reporter lines,

respectively, showing > 75% main bodies with only one bud and ~ 10% main bodies with two or more buds (**Supplementary Fig. 5c, 5g**). Collectively, these results demonstrate the high efficiency and reproducibility of bud formation observed with the Wnt low protocol in various stem cell lines. We have now added these results in the revised manuscript (**Supplementary Fig. 5 and Pages 3-4**).

Due to the overall high percentages of PDX1+ cells (normally 95%-99%) in bulk type clusters, we do not see spontaneous PDX1+ bud formation observed in Wnt med conditions.

- Is there any extracellular matrix coating used during the differentiation? Please include this information as it will be necessary for the reproduction of the protocol.

Response: Thank you for this suggestion. We apply extracellular matrix (ECM) coating for maintenance culture and planar culture during the first four stages of differentiation. Bud induction and subsequent endocrine cell differentiation during the last three stages are conducted under static suspension culture without ECM coating or mounting. We have now emphasized this information in the Methods section in the revised manuscript (**Page 12**).

- Does Pdx1 directly regulate Ephrin signaling and expression? It would be interesting to check the Pdx1 KO line for expression of Ephrin proteins.

Response: Thank you for raising this question. In our budding type Stage 5 clusters, the observation that PDX1+ cells tend to have lower levels of Ephrin proteins (particularly EphrinB2 and EphrinB3) (**Fig. 5 and Supplementary Fig. 15**) may indicate an association between PDX1 and Ephrin expression. Following this suggestion, we have examined the expression of Eph and Ephrin in the PDX1 KO line. We find that all five Eph receptors and three Ephrin ligands are expressed in PDX1 KO Stage 5 clusters (**Supplementary Fig. 17**). However, compared to the differential localization of Eph/Ephrin in endocrine buds and main bodies derived from wildtype cell line (**Supplementary Fig. 15**), we note the differential or polarized expression pattern is diminished in the PDX1 KO clusters (**Supplementary Fig. 17**), suggesting that spatial expression pattern of Eph/Ephrin proteins is associated with the presence or absence of PDX1 and thus the budding morphology. Nevertheless, it requires further in-depth investigation as to whether Eph or Ephrin signaling and expression is directly regulated by PDX1, which could be an interesting work in follow-up studies. We have discussed this result in the revised manuscript (**Page 9**).

- To support further the model and connect with the literature of budding morphogenesis that the authors discuss, it would be interesting to check how the bud “connects” to the main body. Please stain for adhesion molecules e.g., E-cadherin-integrin molecules-EMT markers and cell polarity markers to get a much clear insight into the budding process.

Response: Thank you for raising this interesting question. Following this suggestion, we have examined adhesion molecules, EMT markers and cell polarity markers in our developing cell aggregates. These include E-cadherin, integrin (ITGA1), beta-catenin, collagen (COL4A1/A2), ZO-1 (marking tight junctions), N-cadherin, Vimentin and THY1. We found strikingly elevated expression of adhesion molecules E-cadherin and beta-catenin in the NEUROD1+ endocrine buds (**Supplementary Fig. 18**), indicating an “epithelial” phenotype of bud cells and tight cell-cell contacts via adheren junctions within the bud compartment. COL4A1/A2, a major component of intra-islet basement membrane proteins, was only expressed in endocrine buds (**Supplementary Fig. 18**), which is in line with its abundant presence in human islets (PMID: 32894318). By contrast, higher levels of “mesenchymal” and fibroblast phenotype markers Vimentin and THY1 were expressed in the main bodies (**Supplementary Fig. 13**), again revealing different cell types of buds and main bodies. The expression of EMT markers led us to hypothesize whether PDX1+ cell budding morphogenesis could be mediated by an EMT mechanism. However, our prior experiments of screening various pathways revealed that the EMT-related TGFβ signaling does not affect PDX1+ cell

clustering (**Supplementary Fig. 14**), arguing against an involvement of EMT mechanism in the budding process. We did not detect expression of ITGA1, ZO-1 or N-Cadherin in the clusters, and none of these proteins were found to be enriched in the interface (i.e., boundary region) between buds and main bodies (**Supplementary Fig. 18**). Thus, it remains ambiguous how the bud connects to the main body and requires further investigation in follow-up studies. We have now included these results and relevant discussions in the revised manuscript (**Page 9**).

Minor points:

- Assess directly the PDX1 proliferation rate (e.g., EdU incorporation and FACS) during bud formation.

Response: Thank you for this suggestion. We have examined Ki67 expression in PDX1+ cells by flow cytometry to directly assess PDX1+ cell proliferation rate during the process of bud formation. The data show that proliferative PDX1+ cells (% Ki67+/PDX1+) account for $2.9\% \pm 0.1\%$ of total cell population (Figure 2C) and $8.3\% \pm 1.4\%$ of PDX1+ cell population in S4D6 clusters after aggregation (**Supplementary Fig. 7**). The PDX1+ cell proliferation rate (% Ki67+/PDX1+) is further decreased to $1.1\% \pm 0.03\%$ of total cell population (Figure 2C) and $2.7\% \pm 0.8\%$ of PDX1+ cell population in S5D3 clusters (**Supplementary Fig. 7**). These new results support the notion that formation of PDX1+ buds is unlikely due to local expansion of proliferative PDX1+ cells. We have now included these data in **Fig. 2c** and **Supplementary Fig. 7** in the revised manuscript.

- Some of the data not shown parts would be informative to be included for example:

1. Wnt target genes
2. The screen of the individual components of stage 7 media
3. The data on the screen of different signaling inhibitors on page 6

Response: Thank you. We agree with this reviewer that including these data would be informative and improves the transparency of our results. Following the suggestion, we have now added these data in new supplementary figures. These include: (1) qPCR 96-well plate array data showing differential Wnt pathway genes by Wnt-med and Wnt-low conditions (**Supplementary Fig. 2**); (2) immunostaining data showing the screen of individual components of stage 7 media on islet architecture (**Supplementary Fig. 9**); and (3) live cell fluorescent imaging data showing the screen of different signaling pathways on PDX1+ cell clustering (**Supplementary Fig. 14**).

- If feasible in Supplementary figure 7 validate ephrin expression by western blot

Response: In Figure S7, we detected Eph and Ephrin expression by immunostaining in order to obtain spatial localization and expression patterns of these proteins in the whole clusters. Antibodies used in this study for targeting human Eph and Ephrin have been validated by western blots by the product manufacturers and in multiple research applications. We have specified the information of antibody validation in both the revised manuscript and the “**Reporting Summary**” document.

- For completeness in the zebrafish references please cite this work for ephrin signaling in liver bud formation: PMID: 27825440 and this reference that first showed bud formation in zebrafish larva: PMID: 12941629

Response: Thank you for suggesting these two important publications; both are very relevant to our findings. We have now cited the two references in our revised manuscript (**Fig. 7a and Page 7**).

- A recent preprint used Rfx6 KO iPSC system in the pancreas context, please discuss in relation to your findings <https://doi.org/10.1101/2023.11.15.567202>

Response: Thank you for bringing this manuscript to our attention. In this preprint, the authors report that RFX6 haploinsufficiency impairs beta cell function using iPSC system harboring a specific RFX6 loss-of-

function protein-truncating variant (PTV) - p.His293LeufsTer7. The lack of RFX6 in their homozygous RFX6 KO cells leads to a significant reduction in PDX1 transcript at mRNA level, reduced induction of PDX1+/NKX6.1+ pancreatic progenitor cells and CHGA+ pro-endocrine cells as well as fails to generate major islet cell types (INS+ cells and GCG+ cells), similar to our observations using a RFX6 complete gene knockout system (**Fig. 4a-4f**). Nevertheless, our study reports some unique observations compared to this preprint paper. These include: (1) identification of PDX1+ cell clustering and budding morphogenesis at early progenitor specification stage, which is not reported in this preprint study; (2) normal presence of PPY+ cells in islet clusters derived from our RFX6 gene complete KO cells, whereas the preprint study shows a lack of PPY+ cells in the implants in vivo (but the authors observed an upregulated PPY transcript at mRNA level in vitro). We speculate that different RFX6 disruption systems and culture formats used by the two studies may contribute to the differences in these independent studies. We have now discussed this preprint paper in relation to our findings in our revised manuscript (**Page 10**).

REVIEWERS' COMMENTS

Reviewer #1 (Remarks to the Author):

The authors have done a good job addressing most comments. The last concerns fig5, The authors attribute differences in gene expression of certain genes between bud and body to the different methods used to measure those genes (rna-seq versus qpcr). However, it is still surprising that selected DEGs(particularly, KRT18) seem highly differentially expressed in the heatmap between body and bud protocol. However, no difference is shown by qPCR. Based on the figure legend, they are validating the rnaseq results with qpcr...

Reviewer #2 (Remarks to the Author):

In the revised version of this manuscript the authors have satisfactorily responded to all the points raised by this referee.
The new set of experiments included has greatly improved the work.

Reviewer #3 (Remarks to the Author):

The authors have addressed all my comments and the manuscript has greatly improved in terms of data clarity and reproducibility.
One small notice in Supplementary Figure 18 VIM and THY1 are mentioned in the figure legend but pictures are not shown. Please adjust accordingly.

Reviewer #1 (Remarks to the Author):

The authors have done a good job addressing most comments. The last concerns fig5, the authors attribute differences in gene expression of certain genes between bud and body to the different methods used to measure those genes (rna-seq versus qpcr). However, it is still surprising that selected DEGs (particularly, KRT18) seem highly differentially expressed in the heatmap between body and bud protocol. However, no difference is shown by qPCR. Based on the figure legend, they are validating the rna-seq results with qpcr...

Response: We thank the reviewer for their positive assessment of our studies and for the remaining comment on *KRT18* expressions by RNA-seq (Fig. 5f) and qPCR (Fig. 5j). The RNA-seq data ($\sim\log_2$ scale) of body and bud are 8.84 ± 0.06 versus 8.27 ± 0.10 (t-test $p = 0.01$), and their z-scores (post row z-score scaling) make the difference seem striking on heatmaps. The expressions of *KRT18* in qPCR are 0.32 ± 0.09 in body and 0.15 ± 0.03 in bud (t-test $p = 0.07$), illustrating \sim half of the mean expression values in buds compared to bodies and t-test p value approaching statistical significance. Thus, both techniques show that body cells tend to express a higher level of *KRT18* transcripts relative to bud cells. Furthermore, of the 29 selected differential genes as examined with qPCR (Fig. 5i-5l), 24 genes (82.7%) are cross-validated and show consistent results between RNA-seq and qPCR, demonstrating that qPCR assay is a reliable method to validate RNA-seq data.

Reviewer #2 (Remarks to the Author):

In the revised version of this manuscript the authors have satisfactorily responded to all the points raised by this referee. The new set of experiments included has greatly improved the work.

Response: We thank the reviewer again for their time reassessing our revised manuscript and for their supportive comments on our work.

Reviewer #3 (Remarks to the Author):

The authors have addressed all my comments and the manuscript has greatly improved in terms of data clarity and reproducibility. One small notice in Supplementary Figure 18 VIM and THY1 are mentioned in the figure legend but pictures are not shown. Please adjust accordingly.

Response: Thank you very much for noticing this mistake. Immunostaining data of VIM and THY1 are actually shown in Supplementary Fig. 13 instead of Supplementary Fig. 18. We have now corrected this error in our revised manuscript. We thank the reviewer again for their careful and positive assessment of our studies.